# Interaction of human keratinocytes and nerve fiber terminals at the neuro-cutaneous unit

Christoph Erbacher[1], Sebastian Britz[2], Philine Dinkel[1†], Thomas Klein[1], Markus Sauer[3], Christian Stigloher[2], Nurcan Üçeyler[1]*

[1]Department of Neurology, University Hospital of Würzburg, Würzburg, Germany; [2]Imaging Core Facility, Biocenter, University of Würzburg, Würzburg, Germany; [3]Department of Biotechnology and Biophysics, University of Würzburg, Würzburg, Germany

**Abstract** Traditionally, peripheral sensory neurons are assumed as the exclusive transducers of external stimuli. Current research moves epidermal keratinocytes into focus as sensors and transmitters of nociceptive and non-nociceptive sensations, tightly interacting with intraepidermal nerve fibers at the neuro-cutaneous unit. In animal models, epidermal cells establish close contacts and ensheath sensory neurites. However, ultrastructural morphological and mechanistic data examining the human keratinocyte-nerve fiber interface are sparse. We investigated this exact interface in human skin applying super-resolution array tomography, expansion microscopy, and structured illumination microscopy. We show keratinocyte ensheathment of afferents and adjacent connexin 43 contacts in native skin and have applied a pipeline based on expansion microscopy to quantify these parameter in skin sections of healthy participants versus patients with small fiber neuropathy. We further derived a fully human co-culture system, visualizing ensheathment and connexin 43 plaques in vitro. Unraveling human intraepidermal nerve fiber ensheathment and potential interaction sites advances research at the neuro-cutaneous unit. These findings are crucial on the way to decipher the mechanisms of cutaneous nociception.

*For correspondence: ueceyler_n@ukw.de

Present address: †Institute of Clinical Genetics, Technical University Dresden, Dresden, Germany

Competing interest: The authors declare that no competing interests exist.

## Editor's evaluation

In this important study, Erbacher et al. have used new techniques to explore the neuro-cutaneous structures of human epidermis. Human skin is less studied than rodent skin because it presents challenges in obtaining samples and finding excellent immunohistological labels. Here, the authors have employed expansion microscopy and super-resolution array tomography for histological studies and have developed a human keratinocyte and human iPSC-derived sensory neuron co-culture to provide in vitro data from live cells. Together, the data are compelling, and demonstrate that human axons tunnel through keratinocytes where they form gap junctions that allow for direct cellular communication.

## Introduction

Impairment of the thinly-melinated A-delta and unmyelinated C-nerve fibers may underlie small nerve fiber pathology observed in patients with peripheral (*Birklein, 2005*; *Lacomis, 2002*; *Üçeyler et al., 2013*) and central nervous system diseases (*Nolano et al., 2008*; *Weis et al., 2011*). Cutaneous nerve fiber degeneration and sensitization are hallmarks of small fiber pathology; however, the underlying pathomechanisms are unclear (*Üçeyler, 2016*). The impact of skin cells on nociceptive and

non-nociceptive stimulus detection is increasingly recognized (*Lumpkin and Caterina, 2007*; *Stucky and Mikesell, 2021*).

Physiologically, keratinocytes are the predominant cell type in the epidermis and actively participate in sensory signal transduction and nociception at the neuro-cutaneous unit (NCU). In animal in vitro cell culture models, selective thermal, chemical, or mechanical keratinocyte stimulation led to activation of co-cultured peripheral neurons (*Klusch et al., 2013*; *Mandadi et al., 2009*; *Sondersorg et al., 2014*). Using animal models, nociceptive behavior was induced in mice selectively expressing transient receptor potential vanilloid 1 (TRPV1) in keratinocytes after capsaicin treatment (*Pang et al., 2015*). Mice expressing channelrhodopsin-2 in keratinocytes also displayed pain behavior and intraepidermal nerve fiber (IENF) derived evoked nerve fiber action potentials during laser stimulation (*Baumbauer et al., 2015*). Further, knockout of the mechanoreceptor PIEZO1 in keratinocytes attenuated both innocuous and noxious touch sensation in mice (*Mikesell et al., 2022*).

For underlying functional stimulus transduction of keratinocytes and IENF, signaling molecules such as ATP are increasingly recognized (*Mandadi et al., 2009*; *Moehring et al., 2018*). Hemichannels or gap junctions formed by connexins and pannexins, or vesicular transport may conduct ATP signaling toward afferent nerve fibers (*Barr et al., 2013*; *Maruyama et al., 2018*; *Sondersorg et al., 2014*).

Connexin 43 (Cx43) pores are well established as a major signaling route for keratinocyte-keratinocyte communication (*Tsutsumi et al., 2009*) and potentially transduce external stimuli likewise toward afferents. Signaling might also happen via specialized synapse-like connections to IENF (*Talagas et al., 2020a*). In *Danio rerio* and *Drosophila* models, nerve endings are frequently ensheathed by epidermal cells (*Jiang et al., 2019*; *O'Brien et al., 2012*) and tunneling of fibers through keratinocytes in human skin has recently been shown via confocal microscopy (*Talagas et al., 2020b*). However, the exact mechanisms and mode of signal transduction at the NCU remain elusive.

In an embryonic stem-cell-derived 2D human cell culture model, physical contacts between sensory neurons and keratinocytes were observed hinting to close coupling (*Krishnan-Kutty et al., 2017*). Still, direct and systematic information on ensheathment of human IENF is scarce and ultrastructural architecture or molecular processes remain obscure. Deciphering these contact zones in the human system may profoundly change the understanding of somatosensory processing in health and disease. Ultimately, altered keratinocyte signal molecule release or dysfunctional signaling sites may contribute to cutaneous pain perception (*Talagas et al., 2018*), which could open novel avenues for neuroprotective, regenerative, and analgesic treatment of patients with small fiber pathology (*Keppel Hesselink et al., 2017*).

We aimed at studying exactly these contact zones between keratinocytes and IENF at the NCU in the human system to gain insights on ultrastructure and potential crosstalk in the epidermis. A correlative light and electron microscopy (CLEM) approach via super-resolution array tomography (srAT) in high-pressure frozen, freeze substituted samples (*Markert et al., 2017*) and expansion microscopy (ExM) (*Tillberg et al., 2016*) in diagnostic paraformaldehyde-fixed tissue sections revealed ensheathment and pore protein Cx43 plaques in native human skin. Both phenomena were also present in vitro at the NCU in a fully human keratinocyte and sensory neuron co-culture system which we succeeded to establish. We propose a crucial role of nerve fiber ensheathment and Cx43-based keratinocyte-nerve fiber contacts in small fiber pathology and neuropathic pain pathophysiology widening the scope of somatosensory processing to non-neuronal cells.

## Results

### Human intraepidermal nerve fiber segments are engulfed by keratinocytes

While many IENF were passing between neighboring keratinocytes, srAT revealed IENF ensheathed by keratinocytes in skin samples of all three healthy subjects. This tunneling of fibers was observed both in the basal and upper epidermal layers for several consecutive 100-nm-thin sections indicated via PGP9.5 labeling (*Figure 1A*). IENF intersected closely to either the lateral, posterior, or anterior boundary of the respective keratinocyte and was engulfed by the respective cell for several μm (*Figure 1A*, *Figure 1—video 1*). For advanced tracing of an IENF through the epidermis, a representative site was reconstructed (*Figure 1B*, *Figure 1—video 2*; source images and computed model available at https://zenodo.org/records/6090262). Whilst a major part of the respective nerve fiber

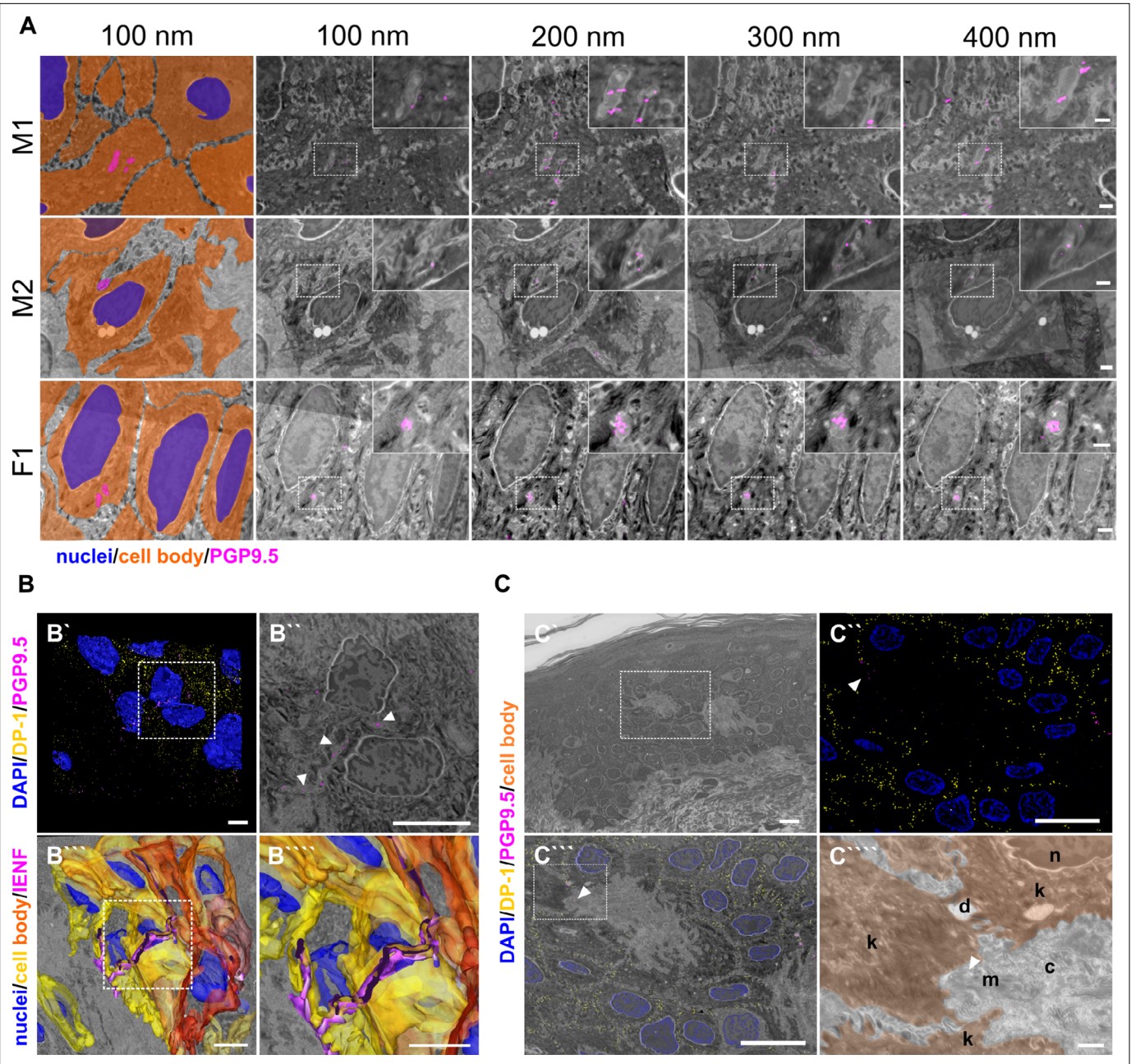

**Figure 1.** Epidermal nerve fiber ensheathment and srAT functionality. (**A**) Ensheathment of IENF by keratinocytes. Nerve fibers projecting within keratinocytes in skin punch biopsy samples of two male (M1, M2) and one female subject (F1). First tile shows keratinocyte cell bodies (orange), nuclei (blue), and fiber (magenta) in pseudo color. Each row represents four consecutive sections with 100 nm thickness of correlated images, with PGP9.5 labeling for IENF (magenta), while dashed insets show higher magnification of the ROI in inlay. See also *Figure 1—video 1*. Scale bar: 1 µm, magnified insets: 500 nm. (**B**) 3D reconstruction of IENF processes traversing between and within keratinocytes. (**B`**) 3D visualization of fluorescence signal from srAT approach, white rectangle indicates area in B``. PGP9.5 (magenta) labeled nerve fiber processes between and in keratinocytes in close apposition to nuclei (blue). DP-1 (yellow) marks intercellular desmosomal junctions as keratinocyte cell boundaries. (**B``**) Single plane with overlay of PGP9.5 signal and EM. (**B```**) Extrapolation of IENF trajectory in 3D, based on IF signal and EM ultrastructure with fiber (magenta), keratinocyte cell bodies (yellow-orange), and nuclei (blue); see also *Figure 1—video 2*. Scale bars: 5 µm. (**C**) Preservation of antigenicity and cellular structure in LR-White embedded epidermal tissue with overview area from SEM (**C`**). Scale bar: 10 µm. (**C``**) SIM image of 100 nm skin section with DP-1 (yellow), PGP9.5 (magenta), and DAPI (blue) labeling. Arrowhead indicates PGP9.5-positive IENF processes. Scale bar: 10 µm. (**C```**) Correlated SIM and SEM image from dashed rectangle in A. Scale bar: 10 µm. (**C````**) Inset of c showing subcellular preservation of collagen fibers (c), desmosomes (de), keratinocytes (k), mitochondria (m), and nucleus (n). Arrowhead indicates IENF processes also observed via IF in a. Scale bar 1 µm. Abbreviations: DP-1, desmoplakin 1; IENF, intraepidermal nerve

*Figure 1 continued on next page*

*Figure 1 continued*

fiber; IF, immunofluorescence; LR-White, London Resin-White; PGP9.5, protein gene product-9.5; SEM, scanning electron microscopy; SIM, structured illumination microscopy.

The online version of this article includes the following video and figure supplement(s) for figure 1:

**Figure supplement 1.** CLEM principle and labeling verification.

**Figure supplement 2.** Utility of srAT for tracing skin cells.

**Figure 1—video 1.** srAT with serial 100 nm sections with one frame per second.

https://elifesciences.org/articles/77761/figures#fig1video1

**Figure 1—video 2.** srAT with serial 100 nm SEM sections and 3D interpolation of keratinocyte nuclei (blue), keratinocyte cell bodies (yellow-orange), and IENF (magenta).

https://elifesciences.org/articles/77761/figures#fig1video2

grew in close contact to and in between keratinocytes, a substantial portion tunneled through one basal keratinocyte. Specificity of antibodies was examined by tracking the fluorophore signal within consecutive slices and negative control via omission of the primary antibody (*Figure 1—figure supplement 1*). High pressure freezing and freeze substitution followed by LR-White embedding preserved the ultrastructure and antigenicity of the human skin tissue as illustrated by identification of collagen fibrils, desmosomes, nuclei, and mitochondria (*Figure 1C*).

The combination of cellular markers such as PGP9.5 or S100β with information on cell morphology from EM scans also allowed tracing of further cell populations via srAT in human skin, including Langerhans cells and dermal Schwann cells (*Figure 1—figure supplement 2*).

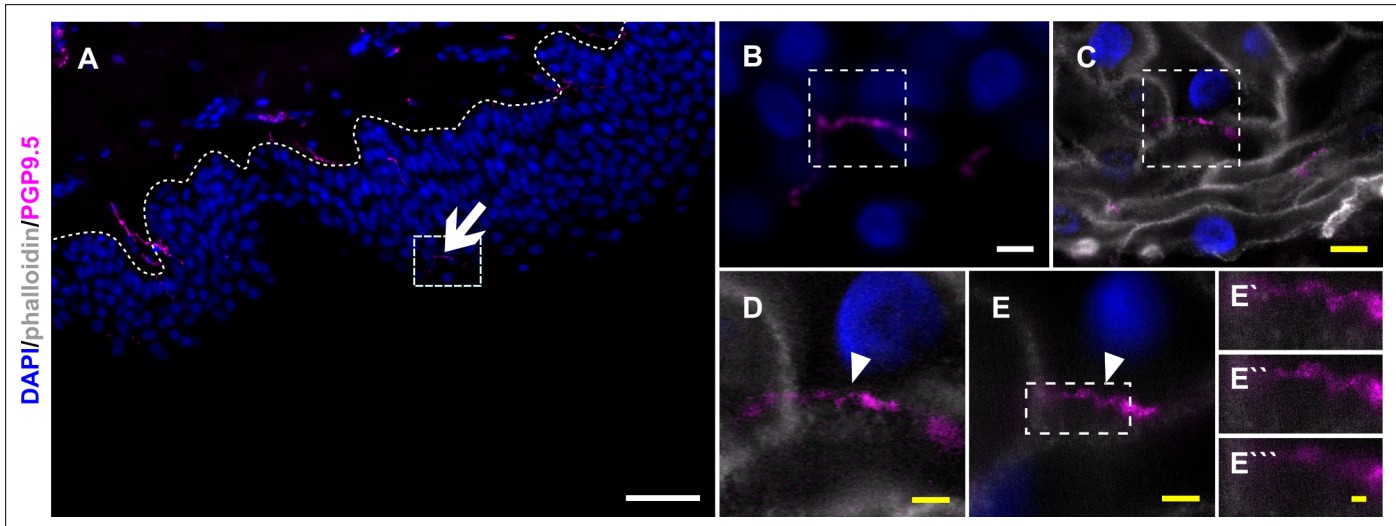

**Figure 2.** Nerve fiber ensheathment in expanded human skin tissue. (**A**) Overview of skin section prior to expansion with PGP9.5-positive IENF (magenta) and nuclear DAPI (blue) signal; dotted line marks epidermis-dermis border and arrow indicates IENF. White rectangle highlights inset enlarged in B. (**B**) Enlarged area prior to expansion and (**C**) matched area post-expansion at same magnification with addition of actin marker phalloidin (grey), white rectangles mark inset area of D and E. (**D**) Enlarged area containing IENF at 20 x magnification and (**E**) at 63 x magnification with arrowheads indicating IENF passing through keratinocyte. Inset in E marks enlarged area in E`-E```. (**E`**) shows z-plane prior to E, (**E``**) same z-plane as E, and E``` z-plane step after E. White scale bars indicate pre-expansion state, yellow scale bars were corrected for expansion factor. Scale bars: 50 µm (**A**), 5 µm (**B, C**), 2 µm (d, e), 500 nm (**E```**). Z-step size of 1.2 µm translates to approximate 276 nm in expanded gel. Abbreviations: DAPI, 4',6-diamidino-2-phenylindole; IENF, intraepidermal nerve fiber; PGP9.5, protein gene product-9.5.

The online version of this article includes the following video and figure supplement(s) for figure 2:

**Figure supplement 1.** Assumed TRPV1 localization in expanded skin tissue section.

**Figure 2—video 1.** Ensheathment and Cx43 plaque in expanded human epidermis.

https://elifesciences.org/articles/77761/figures#fig2video1

## Nerve fiber ensheathment can be visualized by ExM in diagnostic skin samples

To visualize ensheathed nerve fibers also in thicker, diagnostically used skin punch biopsy sections, we applied ExM allowing super-resolution imaging with epifluorescence microscope setups. Expansion factor of samples fell between 4.3 and 4.4 x and showed isotropic epidermal expansion as documented via pre- and post-expansion acquired images (*Figure 2*). Actin filaments were used as a marker to outline epidermal cell bodies and IENF entry and exit points, while cytoplasmic PGP9.5 identified IENF. Nerve fibers were traced via ExM for their course through the epidermis, being ensheathed over several μm (*Figure 2—video 1*). To discriminate IENF subtypes, we applied immunoreaction against TRPV1, however, we could not extract IENF-specific co-localization, also after expansion (*Figure 2—figure supplement 1*).

## Cx43 plaques as potential keratinocyte-nerve fiber communication sites

Connexin hexamers can form hemichannels acting as small pores. In open state, small molecules can pass and be released from the cell, which was already shown for ATP and Cx43 (*Weber et al., 2004*). This may allow purinergic signaling toward neighboring cells or nerve terminals in close proximity. For srAT, Cx43 labeling was assumed a *bona fide* signal, if ≥2 consecutive sections showed fluorescent staining, translating to 200–400 nm. These clusters were mostly found at keratinocyte-keratinocyte contact zones (*Figure 3A*); however, distinct Cx43 plaques were also identified in direct proximity to single IENF when growing between keratinocytes (*Figure 3B*).

In analogy to nerve fiber ensheathment, we investigated Cx43 accumulations also in expanded diagnostic skin samples. In pre-expansion state, the attribution of single Cx43 plaques to specific sites between keratinocytes or toward IENF was hardly possible, due to the compact structure of the epidermis. However, after expansion, specific Cx43-positive accumulations in direct contact to PGP9.5-positive IENF could be identified (*Figure 4*; see also *Figure 2—video 1*).

## Analysis of ensheathment and Cx43 in diagnostic skin samples

We established an imaging and segmentation pipeline of 4 x ExM to analyze IENF ensheathment and Cx43 plaques in clinical skin sections (*Figure 5A*). We compared sizes of epidermal nuclei via Cell-Profiler as an alternative to manual ROI alignment for quality control, which showed a median expansion factor of 3.83±0.36 (*Figure 5B*, *Figure 5—source data 1*). After segmentation, the pixel count of PGP9.5 without WGA co-localization, as a correlate for ensheathment was 65.05% ± 14.51% for control samples and 50.82% ± 28.03% for SFN and did not differ between both groups (*Figure 5C*). Accordingly, the co-localization overlap between PGP9.5 and Cx43 signal was 0.53% ± 0.49% (for controls) and 0.96%±2.16% (for SFN) and did not differ between groups (*Figure 5D*).

## Neurites establish contacts to keratinocytes in fully human co-culture system

Sensory neurons and keratinocytes each formed clusters after seeding into two-comparted chambers (*Figure 6A–C*). After barrier removal, neurites actively grew toward keratinocytes and established contacts within few days (*Figure 6C and D*, *Figure 6—video 1*). Neurite-keratinocyte contacts were apparent, both in conditioned neuronal medium and keratinocyte medium. However, keratinocytes underwent terminal differentiation in neuronal medium, while predominantly maintaining a basal state in keratinocyte medium (*Figure 6—figure supplement 1*). Keratinocyte cell lines and iPSC cell line used for neuronal differentiation were free of mycoplasma and basal keratinocytes were characterized by morphology, positive cytokeratin 14 labeling, and negative cytokeratin 10 labeling (*Figure 6—figure supplement 2*).

## Ensheathment and Cx43 complexes are present in a fully human co-culture model

To distinguish neurite versus keratinocyte membrane, we specifically labeled sensory neurons via cholera toxin subunit B (Ctx) targeting the ganglioside monosialotetrahexosylganglioside 1 (GM1) (*Dederen et al., 1994*; *Tong et al., 1999*). Conversely, the membrane of keratinocytes was targeted by WGA (*Belleudi et al., 2011*; *Watt, 1983*). We identified neurite-keratinocyte contacts using

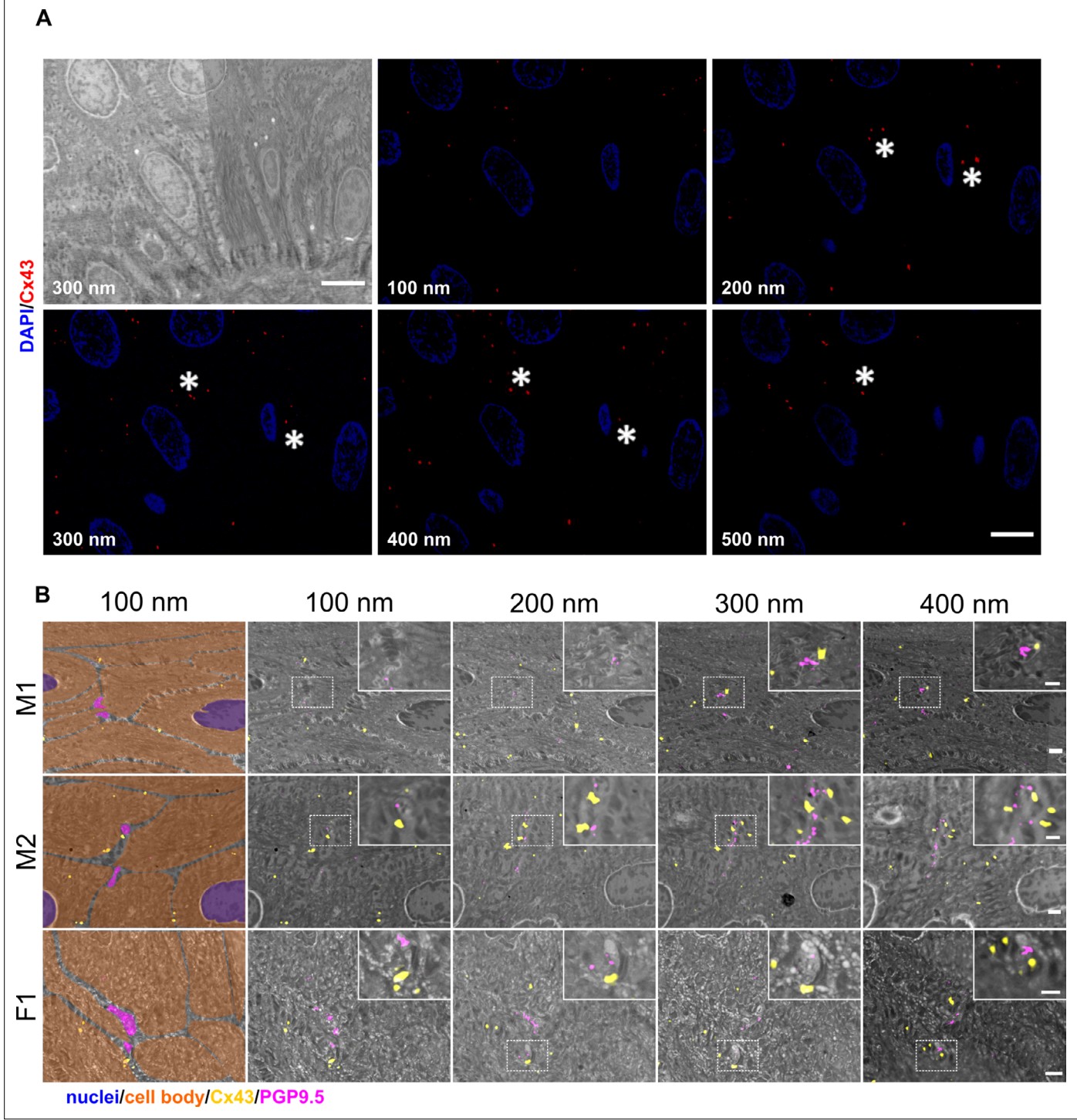

**Figure 3.** Identification of Cx43 plaques via srAT. (**A**) Tracking of Cx43 plaques in epidermal layers. First panel illustrates SEM overview of epidermal layers corresponding to five consecutive sections of IF images showing Cx43 signal (red) and nuclei (blue). Asterisks mark examples of traced Cx43 plaques. (**B**) Cx43 plaques at keratinocyte-nerve fiber close contact sites. Nerve fibers processing between keratinocytes in skin samples of two male subjects (M1, M2) and one female subject (F1). Each row represents four consecutive sections of 100 nm thickness. First tile shows keratinocyte cell bodies (orange), nuclei (blue), and nerve fibers (magenta) in pseudo color with Cx43 signal (red). Correlated PGP9.5 labeling (magenta) locates at nerve fibers and Cx43 labeling (yellow) indicates Cx43 plaques. Insets show magnification of contact area. Scale bars: 5 µm (**A**), 1 µm (**B**), magnified insets: 500 nm. Abbreviations: Cx43, connexin 43; IF, Immunofluorescence; PGP9.5, protein gene product-9.5; SEM, scanning electron microscopy.

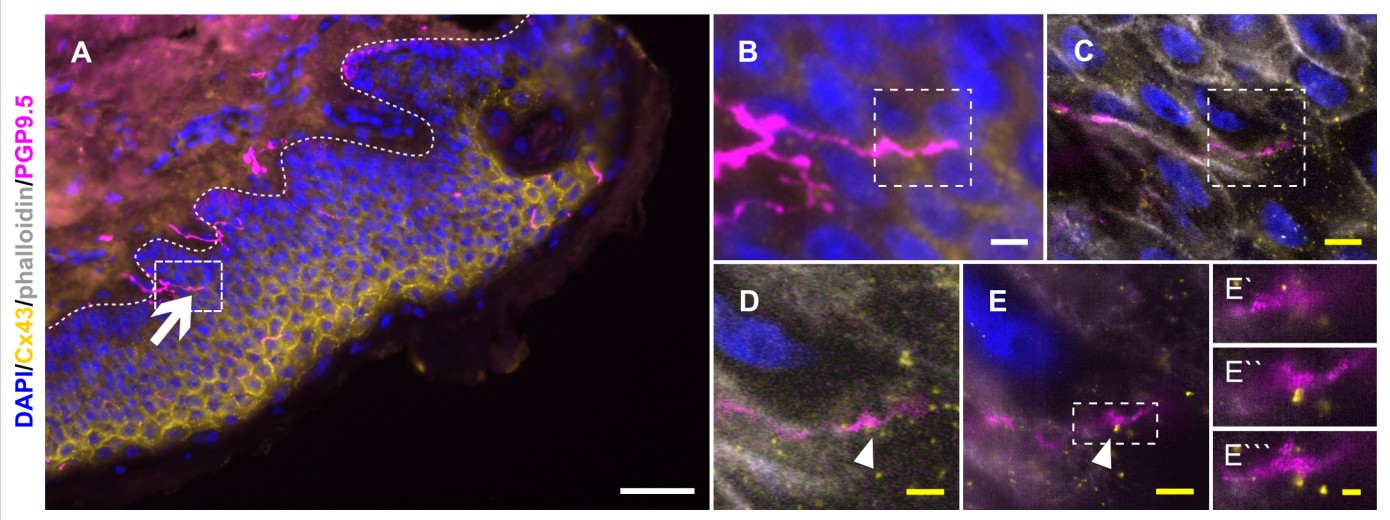

**Figure 4.** Cx43 accumulation at keratinocyte-nerve fiber contact sites in expanded human epidermis. (**A**) Overview of skin section prior to expansion with PGP9.5-labeled IENF (magenta), Cx43 (yellow), and nuclear DAPI (blue); dotted line illustrates epidermis-dermis border and arrow indicates IENF. White rectangle marks inset enlarged in B. (**B**) Enlarged area prior to expansion and (**C**) matched area post-expansion at same magnification with addition of actin marker phalloidin (grey); white rectangles mark inset area of D and E. (**D**) Enlarged IENF area at ×20 magnification and (**E**) at ×63 magnification with arrowheads indicating Cx43 plaque at IENF. Inset in E marks enlarged area in E`-E```. (**E`**) shows z-plane prior to E, (**E``**) same z-plane as E, and E``` z-plane step after E. White scale bars indicate pre-expansion state, yellow scale bars are corrected for expansion factor. Scale bars: 50 µm (**A**), 5 µm (**B, C**), 2 µm (**D, E**), 500 nm (**E`-E```**). Z-step size of 1.2 µm translates to approximate 274 nm in expanded gel. Abbreviations: Cx43, connexin 43; IENF, intraepidermal nerve fiber; PGP9.5, protein gene product-9.5.

confocal microscopy (*Figure 7A and B*) and observed ensheathment via lattice-SIM super-resolution microscopy (*Figure 7C–C``*). Non-ensheathed neurites frequently passed in close proximity and over keratinocytes (*Figure 7D*). Intriguingly, Cx43 labeling revealed Cx43 plaques at those passing sites, distributed over several individual keratinocytes (*Figure 7D–F``*).

We further found neurites growing in a gutter-like structure (*Figure 8A*), merging into the keratinocyte membrane (*Figure 8B*) and observed neurites that establish bouton-like contacts with keratinocytes (*Figure 8C*). We included synaptophysin (SYP) labeling as a marker for small synaptic vesicles, which might serve as another pathway of signal transduction between keratinocytes and IENF (*Talagas et al., 2020a*). In our iPSC-derived neurons, SYP was distributed throughout the cytoplasm and not restricted to the cytoskeleton (Figure 8—figure supplement 1). Conversely, only weak SYP labeling, not associated with neurite contact sites, was present in keratinocytes (*Figure 8A–C*).

## Keratinocytes and neurites show temporarily coupled Ca²⁺ increase

Ca²⁺ imaging of human co-cultured sensory neurons and keratinocytes revealed spontaneous activity in both cell types (4/7 wells). Keratinocytes showed slower and longer Ca²⁺ peaks, while neurites exhibited a sharp rise (*Figure 9*, *Figure 9—video 1*). In three wells with neurites passing along keratinocytes, an increase of keratinocyte Ca²⁺ preceded neurite Ca²⁺ peaks. Together, this indicates functional activity in our co-culture model, possibly coupled between both cell types.

## Discussion

We have investigated the NCU in human skin and provide evidence for nerve fiber ensheathment by keratinocytes and Cx43 contact sites between keratinocytes and IENF. These findings may profoundly change the view on the role neuronal and non-neuronal cells play in the development and maintenance of neuropathy and neuropathic pain.

Ensheathment was previously described in model organisms with mono- or double-layered epidermis such as *Drosophila* and *Danio rerio* (*Han et al., 2012*; *O'Brien et al., 2012*). In multi-layered mammalian and human skin, only sparse data exist from early EM studies reporting conflicting observations. Given the small diameter of IENF (≤1 µm) within the dense tissue of the epidermis, application

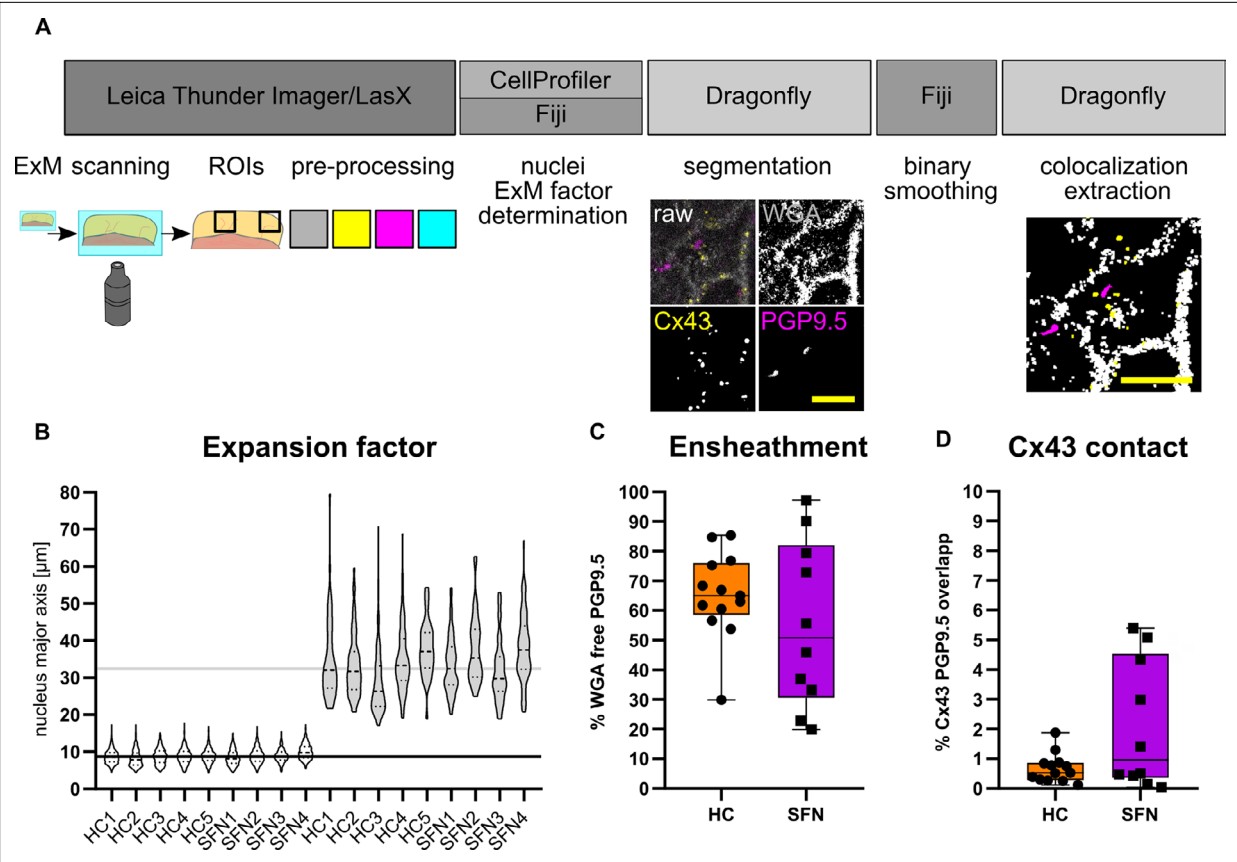

**Figure 5.** Quantitative assessment of morphological parameters at the NCU. (**A**) Workflow for ensheathment and Cx43 quantification (see also *Figure 5—video 1*). (**B**) Magnitude of skin tissue expansion across investigated sections via comparison of epidermal nuclei. Median nucleus size between samples pre-ExM (8.73±0.54 μm) and post ExM (32.46±3.54 μm) denotes a median ExM factor of 3.83±0.36. (**C**) Extracted fiber ensheathment ratio (PGP9.5-positive px without WGA pixel co-localization divided by total PGP9.5 px count) in HC versus SFN patients. (**C**) Extracted fiber ratio (total PGP9.5 px versus PGP9.5 non WGA overlap px) in HC versus SFN patients (n=13 HC and 10 HC ROIs from n=5 HC and n=4 SFN samples; p=0.41; shown as box & whiskers plot with minimum and maximum). (**D**) Cx43 contact ratio (PGP9.5 px with Cx43-WGA double positive px divided by total PGP9.5 px) in HC versus SFN (n=13 HC and 10 SFN ROIs from n=5 HC and n=4 SFN samples; p=0.34; shown as box & whiskers plot with minimum and maximum). Expansion factor corrected scale bars (yellow) of exemplary segmentation: 5 μm. See also *Figure 5—source data 2*. Abbreviations: HC, healthy control; Cx43, connexin 43; NCU, neuro-cutaneous unit; PGP9.5, protein gene product 9.5; px, pixel; SFN, small fiber neuropathy; WGA, wheat germ agglutinin.

The online version of this article includes the following video and source data for figure 5:

**Source data 1.** Descriptive data and values for nucleus based calculation of expansion factors.

**Source data 2.** Statistical details and input values for *Figure 5C and D*.

**Figure 5—video 1.** Image processing of quantification pipeline.

https://elifesciences.org/articles/77761/figures#fig5video1

of super-resolution microscopy techniques is inevitably necessary to resolve the exact course of these neurites. Recently, 'tunneling' of IENF within keratinocytes was proposed via confocal microscopy in human skin (*Talagas et al., 2020b*) and via SEM in an heterologous rat-human co-culture model (*Talagas et al., 2020a*). srAT and ExM techniques used in our study fortify these findings at ultra-structural level and ExM opens the avenue for detailed assessment in diagnostically relevant tissue sections. We further identified Cx43 plaques of keratinocytes in close proximity to IENF as potential components of the NCU exactly size-matching similar connexin and innexin plaques (*Agullo-Pascual et al., 2013*; *Markert et al., 2016*; *Taki et al., 2018*).

Keratinocyte-keratinocyte communication via calcium wave propagation and ATP release are canonical functions of Cx43, orchestrating proliferation, wound healing, and inflammatory processes (*Martin et al., 2014*; *Tsutsumi et al., 2009*). It is of note that ATP was also found to be a direct signal

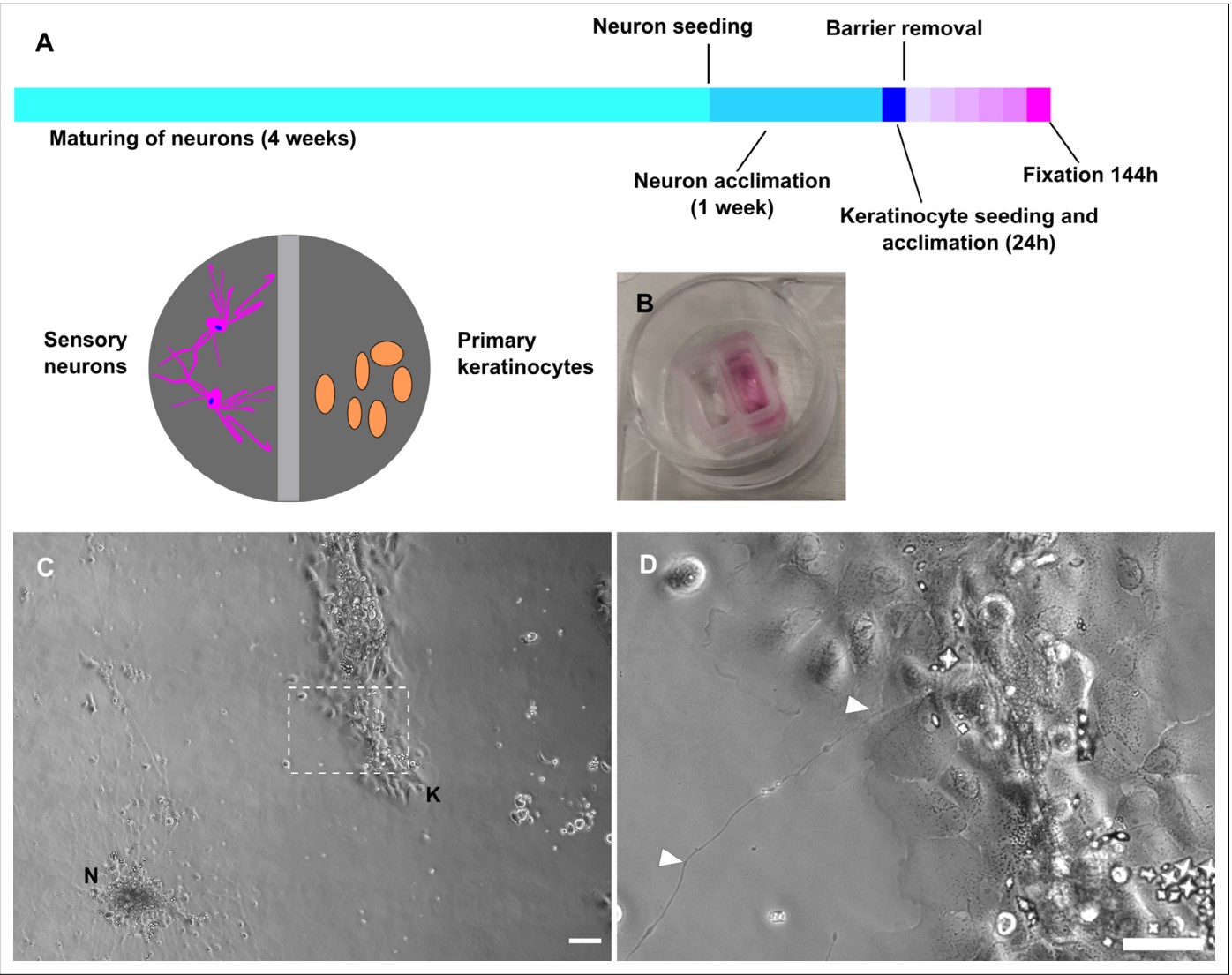

**Figure 6.** Fully human sensory neuron-keratinocyte co-culture model. (**A**) Timeline of culturing protocol and compartment scheme. (**B**) Chamber system. (**C**) Overview of co-culture after 115 h in conditioned neuronal medium with neuronal cluster [N] and keratinocyte colony [K]. Inset (**D**) shows a single neurite in contact with keratinocytes (arrowheads). Co-culture kept in conditioned neuronal medium. Scale bars: 100 µm (**C**), magnified inset: 50 µm (**D**). See also *Figure 6—video 1*. Abbreviations: K, keratinocyte colony; N, neuronal cluster.

The online version of this article includes the following video and figure supplement(s) for figure 6:

**Figure supplement 1.** Comparison of co-culture dependent on media condition.

**Figure supplement 2.** Keratinocyte characterization and mycoplasma screen.

**Figure 6—video 1.** Live imaging of fully human sensory neuron-keratinocyte co-culture with neurite establishing contact to keratinocyte colony.
https://elifesciences.org/articles/77761/figures#fig6video1

transducer from keratinocytes to sensory neurites (*Cook and McCleskey, 2002*; *Sondersorg et al., 2014*).

Interactions at the NCU may have unprecedented implications for a wide range of somatosensory functions in health and disease. In *Drosophila* larvae, a bidirectional guidance mechanism stabilizing existing fibers and limiting fiber arborization was proposed, maintaining sensory receptive fields. In this model, disturbance of ensheathment reduced nocifensive behavior (*Jiang et al., 2019*). In human patients, small fiber pathology is characterized by functional and/or morphological impairment of IENF and is a common finding in a range of neurodegenerative, metabolic, and chronic pain-associated diseases (*Ghasemi and Rajabally, 2020*; *Pittenger et al., 2005*; *Vinik et al., 2001*).

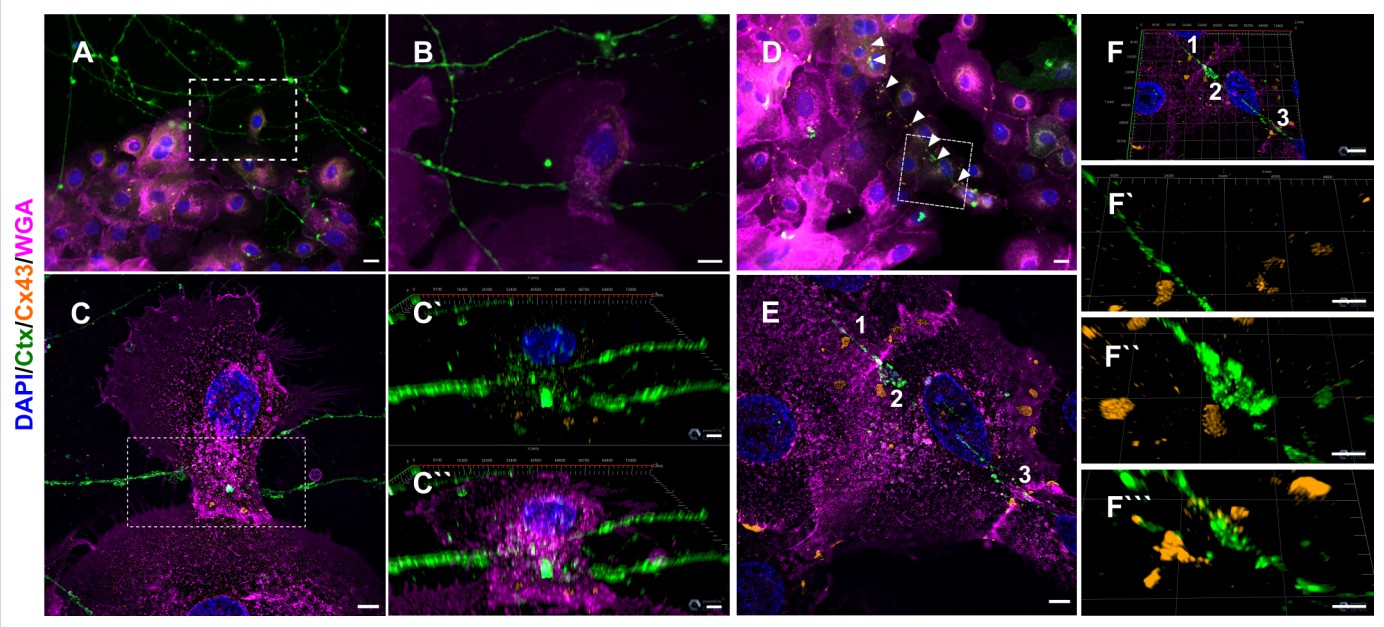

**Figure 7.** Neurite ensheathment and Cx43 plaques in full human co-culture model. (**A**) Confocal overview image of Ctx-positive sensory neurites (green), Cx43 positive (orange) keratinocytes with membrane labeling of keratinocytes via WGA (magenta) and nuclear DAPI (blue). (**B**) Inset of ensheathment area from A. (**C**) Single plane lattice SIM image and respective inset area with 3D visualization of z-stack (2.925 μm depth, 0.196 μm steps) showing nucleus, Cx43, and neurite signal (**C`**) and including WGA (**C``**). (**D**) Confocal overview image of Ctx-positive sensory neurites (green), Cx43-positive (orange) keratinocytes with membrane labeling of keratinocytes via WGA (magenta), and nuclei (blue). Arrowheads indicate Cx43 - neurite contact areas. (**E**) Single plane lattice SIM image and respective inset area with 3D visualization, numbers represent single Cx43 plaques (**F**) of z-stack (2.925 μm depth, 0.196 μm steps). (**F`-F``**) detailed neurite - Cx43 contact areas. Co-culture kept in keratinocyte medium. Scale bars: 20 μm (**A**, **D**), 5 μm (**C-C`**, **E**, **F`-F```**), 1 μm (**F**). Abbreviations: Ctx, cholera toxin subunit B; Cx43, connexin 43; DAPI, 4',6-diamidino-2-phenylindole; SIM, structured illumination microscopy; WGA, wheat germ agglutinin.

Skin punch biopsies are an easily accessible biomaterial of increasingly acknowledged diagnostic value (*Evdokimov et al., 2019*; *Lin et al., 2016*). In a pilot experiment, we saw no differences of ensheathment rate or Cx43 expression in skin samples of patients with SFN compared to healthy controls except for higher variance in the patient cohort. Still, our imaging pipeline provides a valuable starting point for quantitative analysis of morphological parameters at super-resolution. It is applicable for clinical frozen skin sections, for virtually all markers with validated antibodies, and is also applicable in large experimental groups. Intriguingly, new markers for nociceptor sub-classification are emerging via transcriptomic profiling of human DRG neurons, which will enable profiling keratinocyte-IENF interactions based on fiber subtypes (*Bhuiyan et al., 2023*).

These data may help understand the pathophysiology of diseases of the peripheral and central nervous system including dystrophic changes typically found in skin of patients with neuropathies (*Hovaguimian and Gibbons, 2011*).

Handling of large super-resolution data, variability of fluorescence signals across expanded sections, and complex segmentation challenges account for the small cohort sizes as a major limitation of our study. Our multi-stepwise segmentation pipeline may enable more standardized and unbiased extraction of morphology-based parameters, while highly processed images also inherit risks for systematic errors like over- or underestimation. These limitations may be overcome by software guided image acquisition of only skin ROIs containing IENF and more sophisticated deep learning algorithms for downstream 3D segmentation guided by artificial intelligence (*D'Antuono and Pisignano, 2022*; *Stringer et al., 2021*).

It is pivotal to recognize and further investigate the active role of keratinocytes within the NCU. Keratinocytes communicate with IENF via ATP and facilitate normal and nociceptive sensory perception in mechanical and thermal modalities (*Moehring et al., 2018*; *Sadler et al., 2020*). Vesicles (*Maruyama et al., 2018*), pannexins (*Sondersorg et al., 2014*), and connexins (*Barr et al., 2013*) are potential mediators of this ATP release and might be dependent on the evoking stimulus. Our

**eLife** Research article

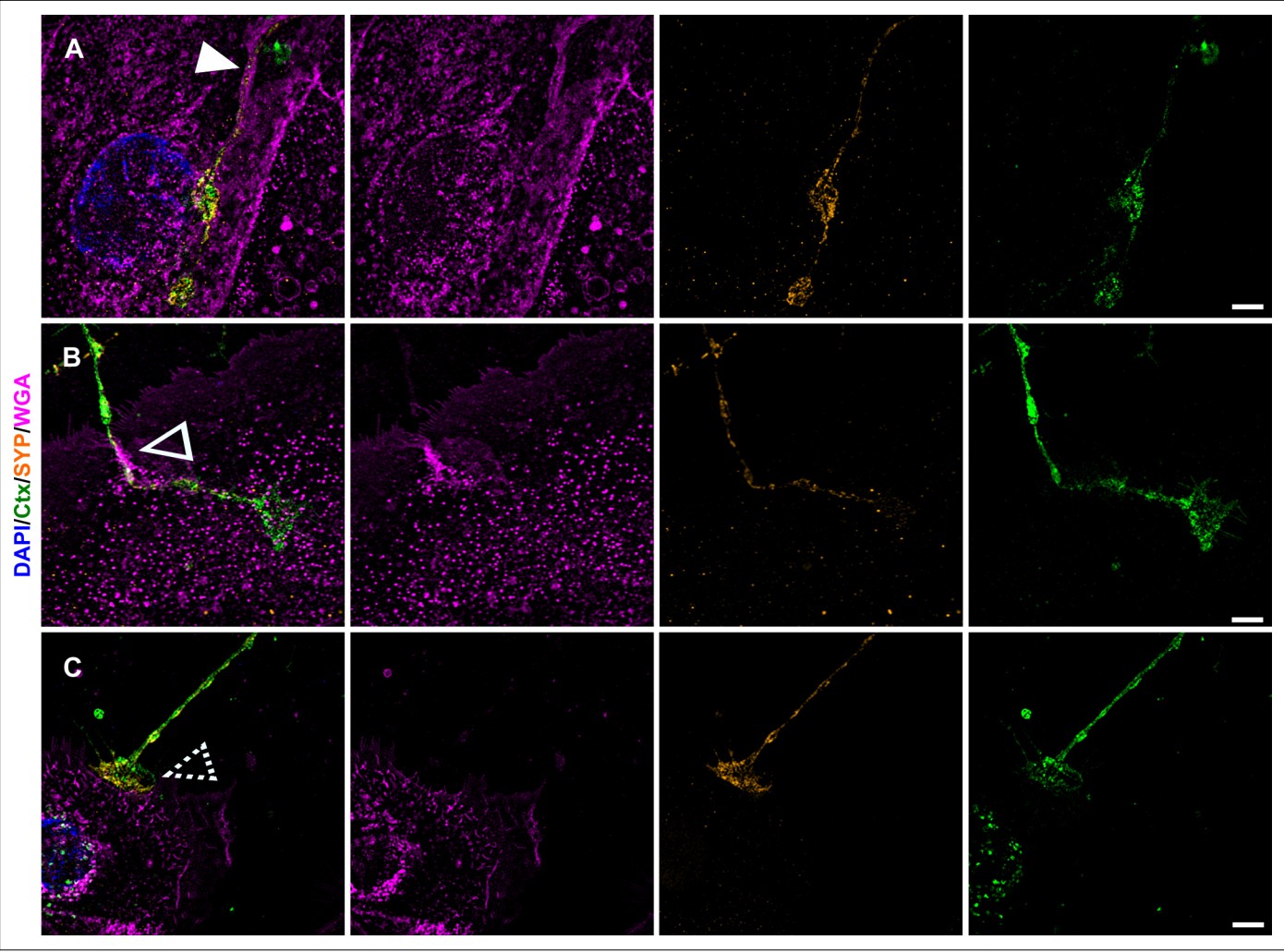

**Figure 8.** Keratinocyte-neurite interactions and synaptic vesicular SYP distribution. Single plane lattice SIM images with overlay of nuclear DAPI (blue), WGA (magenta), SYP (yellow), and Ctx (green) signal as first panel, followed by single channel images of WGA, SYP, and Ctx. Distinct contact sites with gutter like structure (**A**) indicated by filled arrowhead, enwrapping (**B**) indicated via empty arrowhead, and bouton-like contact (**C**) indicated via dashed arrowhead were observed in human keratinocyte-sensory neuron co-culture. SYP signal in A-C is predominantly restricted to neurites with sparse dotted labeling in keratinocytes. Co-culture kept in keratinocyte medium. Scale bars: 5 μm. Abbreviations: Ctx, cholera toxin subunit B; DAPI, 4',6-diamidino-2-phenylindole, SIM, structured illumination microscopy; SYP, synaptophysin; WGA, wheat germ agglutinin.

The online version of this article includes the following figure supplement(s) for figure 8:

**Figure supplement 1.** Neurite outline and synaptic vesicular SYP localization.

observation of Cx43 plaques along the course of IENF in native skin and a human co-culture model substantiates a morphological basis and suggests keratinocyte hemichannels or gap junctions as one potential signaling pathway toward IENF. A single-cell RNA-sequencing approach of human epidermal cells determined 'channel keratinocytes' with upregulated pore and intercellular communication transcripts, for example Cx26 and Cx30 (*Cheng et al., 2018*).

Hemichannel or even gap junctional communication between keratinocytes and IENF might hence not be restricted to Cx43 and differentially organized in varying specialized keratinocytes.

We successfully established a fully human co-culture model of sensory neurons and keratinocytes maintaining viability for at least 6 days and neurites growing toward and interacting with keratinocytes. The 2D culture system reduced the multilayered complexity of native skin, yet conserved ensheathment and Cx43 plaques as hallmarks of the NCU. Embryonic stem-cell-derived human sensory neurons and human keratinocytes were successfully co-cultured before resulting in direct contacts and engulfed neurites (*Krishnan-Kutty et al., 2017*). However, our model uses fibroblast-derived iPSC which can

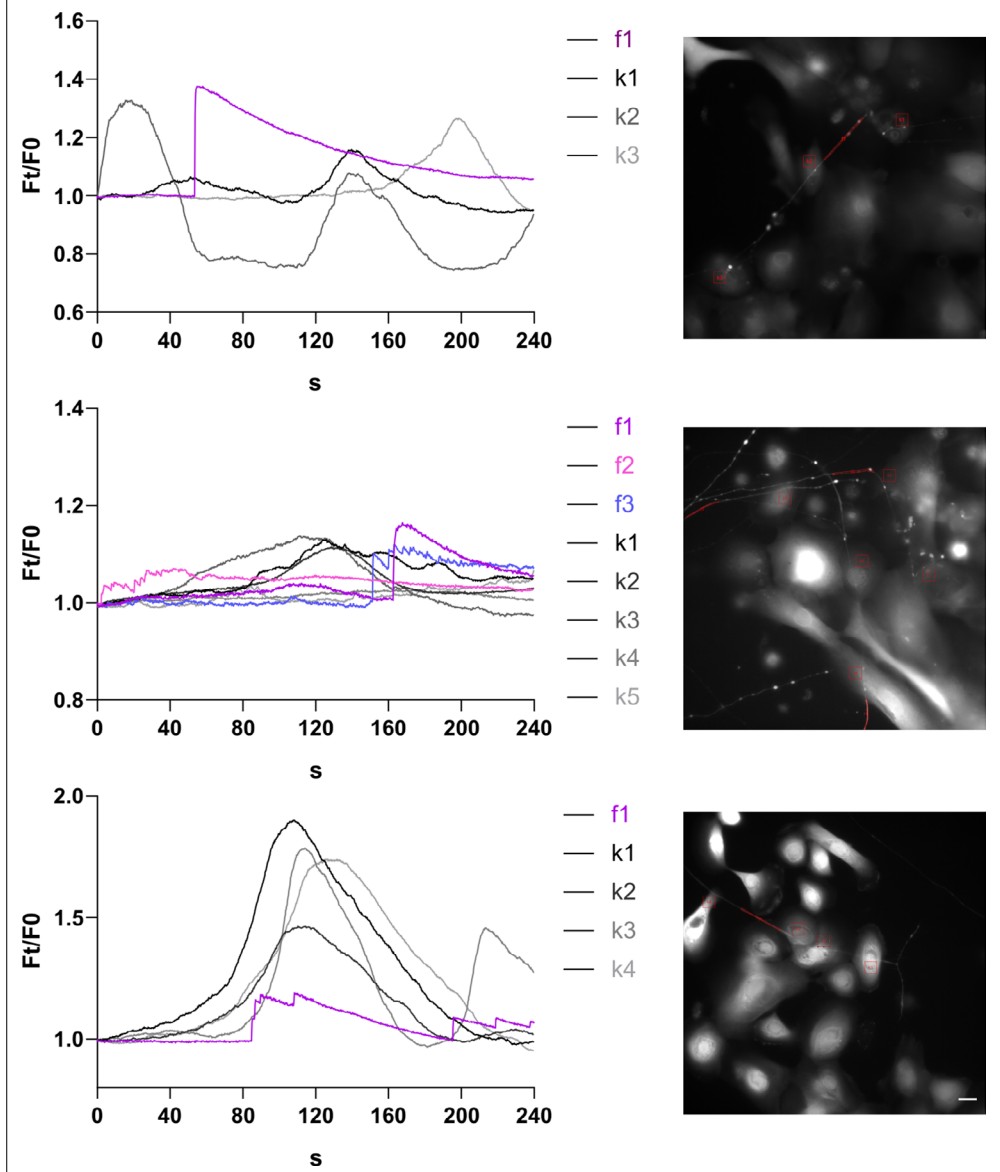

**Figure 9.** Ca²⁺ traces in fully human co-culture systems of sensory neurons and keratinocytes are temporally connected. (Left) Representative Ca²⁺ transients of keratinocytes (black-grey) and neurites (magenta-blue) in contact and (right) imaged area with respective ROIs. Scale bar: 20 μm. Spontaneous activity observed in 3/7 wells. See also *Figure 9—video 1*.

The online version of this article includes the following video for figure 9:

**Figure 9—video 1.** Ca2⁺ imaging of fully human sensory neuron-keratinocyte co-culture.
https://elifesciences.org/articles/77761/figures#fig9video1

be generated from virtually any relevant group of patients. Recently, a heterologous model of rat DRG neurons and human keratinocytes showed neurites passing along a keratinocyte gutter or being ensheathed by keratinocytes, which matches our observations. Additionally, SYP, synaptotagmin, and syntaxin 1 A were successfully labeled together with cytokeratin 6 as a keratinocyte marker and pan-neurofilament as a neurite marker, suggesting en passant synapse-like contacts (*Talagas et al., 2020a*). Further, this study described tunneling fibers and 'gutters' in their heterologous co-culture system, being congruent to our observations in the human keratinocyte- sensory neuron system. However, neurofilaments represent intermediate filaments within the cytoplasm and may not encompass the whole neuronal outline compared to membrane labeling (see *Figure 8—figure supplement*

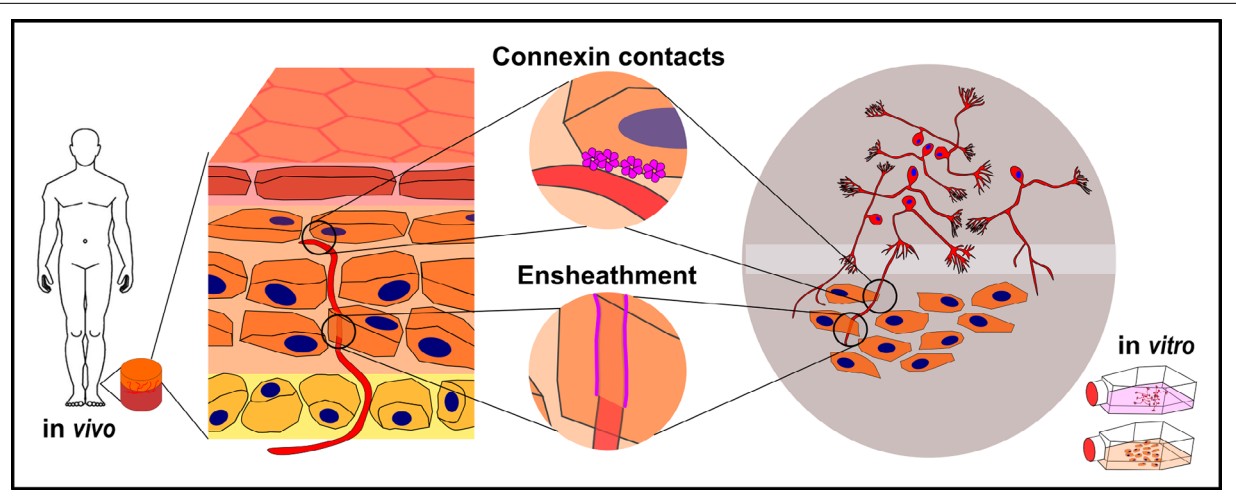

**Figure 10.** Keratinocyte-nerve fiber interactions in human epidermis and 2D model. Proposed connexin contacts are potential transducers of sensory and nociceptive keratinocyte adenosine triphosphate signaling toward intraepidermal nerve fibers. Ensheathment of fibers by keratinocytes may orchestrate nerve fiber outgrowth and stabilization. Both observations are conserved in human 2D co-culture model.

*1*). In accordance with findings from native DRG neurons, our sensory iPSC neurons contained SYP accumulations within the cytoplasm (*Chou et al., 2002*; *Chung et al., 2019*), deeming a membranous labeling necessary to clearly localize SYP in keratinocytes and rule out a neuronal localization within a co-culture approach. In our study, we did not observe specific SYP-positive clusters or vesicles in the cytosol of primary human keratinocytes associated with passing neurites.

Still, electrophysiological activity of neurons in contact with keratinocytes has shown to be attenuated after blockage vesicular secretion of keratinocytes via botulinum neurotoxin type C, hinting toward a physiological role of synaptic release in signal transduction at the NCU (*Talagas et al., 2020a*).

As proof-of-concept for signal transduction between keratinocytes and neurons in our fully human co-culture, we detected spontaneous, time-coupled $Ca^{2+}$ peaks in keratinocytes and neurites. However, our ibidi well system is not ideal for stimulation experiments. Specialized microfluidic chambers with directional neurite outgrowth, coupled with perfusion systems are needed to validate and quantify various modes of keratinocyte-neurite interactions at higher throughput combined with keratinocyte-specific stimulation or antagonists like the gap junction blocker carbenoxolone.

Our data further show that diagnostic interpretation of IENF density in skin punch biopsies solely based on PGP9.5 labeling deserves some caution. PGP9.5 is a cytoplasmic marker that may not be distributed homogenously along the whole fiber and omits the neurites membrane, which will be important for localization and co-localization of involved proteins at super-resolution. Regarding in vitro approaches, our homologous model should minimize variability and potential artifacts of heterologous co-cultures, especially since profound differences in neuronal DRG subpopulations can be observed across mammalian species (*Klein et al., 2021*; *Kupari et al., 2021*; *Shiers et al., 2020*).

## Conclusion

Sophisticated cell culture and animal models along with super-resolution microscopy are barely beginning to unveil the complexity of the NCU. Epidermal keratinocytes show an astonishing set of interactions with sensory IENF including ensheathment and potential electrical and chemical synapse-like contacts to nerve fibers (*Figure 10*) which may have substantial implications for the pathophysiological understanding of neuropathic pain and neuropathies. Our morphological findings underline the significance of keratinocytes in somatosensory and cutaneous nociception and add to the increasing change of textbook knowledge viewing sensory fibers as the sole transducers of environmental stimuli. Expanding investigations toward skin cell impairment in small fiber pathology will help to better understand the underlying mechanisms and open new avenues for targeted treatment.

# Materials and methods

**Key resources table**

| Reagent type (species) or resource | Designation | Source or reference | Identifiers | Additional information |
|---|---|---|---|---|
| Biological samples | Skin sections | This paper | | See *Supplementary file 1* |
| Biological samples | Primary keratinocytes | This paper | | See *Supplementary file 1* |
| Biological samples | iPSC and-differentiated sensory neurons | *Klein et al., 2023* | | see *Supplementary file 1* |
| Antibody | Monoclonal mouse anti-protein gene product 9.5 | AbD serotec, Puchheim, Germany | Cat# 7863–1004 | srAt (1:400); ExM (1:200) |
| Antibody | Monoclonal rabbit anti-S100β | Abcam, Cambridge, UK | Cat# ab52642 | srAt (1:400) |
| Antibody | Mouse anti-Neurofilament marker (pan-neuronal, cocktail) | BioLegend, San Diego, CA, USA | Cat# 837801 | ICC (1:100) |
| Antibody | Polyclonal guinea pig anti-Desmoplakin | Progen, Heidelberg, Germany | Cat# DP-1 | srAt (1:400) |
| Antibody | Polyclonal rabbit anti-protein gene product 9.5 | Zytomed, Berlin, Germany | Cat# 516-3344 | ExM (1:100) |
| Antibody | Polyclonal rabbit anti-Connexin 43 | Sigma Aldrich, St. Louise, MO, USA | Cat# C6219 | ExM (1:100); ICC (1:100); srAT (1:400) |
| Antibody | Polyclonal guinea pig anti-Connexin 43 | alomone labs, Jerusalem, Israel | Cat# ACC-201-GP | ExM (1:100) |
| Antibody | Polyclonal rabbit anti-Synaptophysin | Merck, Darmstadt, Germany | Cat# AB9272 | ICC (1:100) |
| Antibody | Monoclonal mouse anti-Cytokeratin 10 | Abcam, Cambridge, UK | Cat# ab1421 | ICC (1:200) |
| Antibody | Polyclonal guinea pig anti Cytokeratin 14 | Progen, Heidelberg, Germany | Cat# GP14 | ICC (1:200) |
| Antibody | Alexa Fluor 488 polyclonal donkey anti-mouse | Dianova, Hamburg, Germany | Cat# 715-545-150 | ExM (1:200); srAT (1:400) |
| Antibody | Alexa Fluor 488 polyclonal donkey anti-rabbit | Dianova, Hamburg, Germany | Cat# 711-545-152 | ExM (1:200) |
| Antibody | CF568 donkey polyclonal anti-rabbit | Biotium, Fremont, CA, USA | Cat# 20098–1 | ExM (1:200) |
| Antibody | CF568 donkey polyclonal anti-guinea pig | Biotium, Fremont, CA, USA | Car# 20377 | ExM (1:200) |
| Antibody | SeTau-647 anti-Rabbit | Conjugated antibody kindly provided by Prof. Markus Sauer, Department of Biotechnology and Biophysics, University of Würzburg, Germany. | | srAT (1:400) |
| Antibody | CF633 goat polyclonal anti-rabbit | Biotium, Fremont, CA, USA | Cat# 20122–1 | ExM (1:200) |
| Antibody | Cy3 goat polyclonal anti-guinea pig | Dianova, Hamburg, Germany | Cat# 106-165-003 | srAT (1:400) |
| Commercial assay | Fluo8-AM ester | Abcam, Cambridge, UK | Cat# ab142773 | 2 µM |
| Software | ImageJ/FIJI | National Institutes of Health; *Schindelin et al., 2012* | | doi:10.1038/nmeth.2019 |
| Software | ImageJ plugin TRAKEM2 | *Cardona et al., 2012*; *Saalfeld et al., 2012* | | https://github.com/trakem2/TrakEM2 |
| Software | Inkscape | Inkscape project | | https://inkscape.org/?switchlang=en |

*Continued on next page*

*Continued*

| Reagent type (species) or resource | Designation | Source or reference | Identifiers | Additional information |
|---|---|---|---|---|
| Software | GIMP2 | The GIMP Development Team | | https://www.gimp.org/downloads/ |
| Software | IMOD | *Kremer et al., 1996* | | https://bio3d.colorado.edu/imod/ |
| Software | Dragonfly ORS | Dragonfly - a brand of Comet | | https://www.theobjects.com/dragonfly/index.html |
| Software | ImageJ BaSiC plugin | *Peng et al., 2017a*; *Peng et al., 2017b* | | https://github.com/marrlab/BaSiC |
| Software | ImageJ plugin MosaicJ | Swiss Federal Institute of Technology Lausanne | | |
| Other | Actin ExM 546 (phalloidin derivate) | Chrometra, Kortenaken, Belgium | | ExM (33 nM) |
| Other | Cholera Toxin Subunit B (Recombinant), Alexa Fluor 488 Conjugate | Thermo Fisher Scientific, Waltham, ME, USA | | ICC (1:100) |
| Other | Wheat germ agglutinin, Alexa Fluor 647 conjugate | Thermo Fisher Scientific, Waltham, ME, USA | | ICC (1:100) |
| Other | Wheat germ agglutinin, CF633 conjugate | Biotium, Fremont, CA, USA | | ExM (1:100) |

## Participants

Our study was approved by the Würzburg Medical School Ethics committee (#135/15). All study participants were recruited at the Department of Neurology, University Hospital of Würzburg, Germany between 2017 and 2022. Epidemiological parameters of all study participants are summarized in *Supplementary file 1*. For CLEM, we enrolled three healthy volunteers (two men [26 and 28 years] and one woman [33 years]). For ExM and cell culture experiments, we used skin punch biopsy material collected from four patients (3 women and 1 man) with small fiber neuropathy (SFN) and nine healthy controls (eight women and one male) following a standard operation procedure as described previously (*Egenolf et al., 2021*). Since we used non-high-throughput methods and provide proof-of-principle data, our study was restricted to selected cases and sample size estimation was not applicable.

## Biomaterial

Skin punch biopsies were obtained using a commercial punch device (Stiefel GmbH, Offenbach, Germany) and following a standard procedure (*Üçeyler et al., 2010*). For CLEM, all three healthy controls underwent a 2 mm skin punch biopsy from the back at th10 level. For tissue sections subjected to ExM and for deriving cell cultures, we used skin sections obtained via 6 mm skin punch biopsies from the upper thigh and lower leg according to a previously published protocol (*Karl et al., 2019*).

## srAT sample preparation

Biopsies were immediately wetted in freezing solution composed of 20% (w/v) polyvinylpyrrolidon in phosphate buffered saline (PBS; 0.1 M, pH = 7.4) to prevent dehydration.

The epidermal layer was manually dissected from dermal and subdermal compartments of the skin sample and transferred into a type A aluminium specimen carrier (Leica Microsystems, Wetzlar, Germany) with recesses of 200 µm containing polyvinylpyrrolidon and capped with a second carrier without recess (Leica Microsystems). Subsequent high pressure freezing and freeze substitution was applied as described previously (*Markert et al., 2016*). A total of 100 nm serial sections were cut via a histo Jumbo Diamond Knife (DiATOME, Biel, Switzerland) with an ultra-microtome EM UC7 (Leica Microsystems, Wetzlar, Germany). Sections were connected as array by adhesive glue (pattex gel compact, Henkel, Düsseldorf-Holthausen, Germany) mixed with xylene (AppliChem, Darmstadt, Germany) and Spinel Black 47400 pigment (Kremer pigmente, Aichstetten, Germany), which was

**Table 1.** Primary and secondary antibodies and marker with specifications.

| Primary antibodies/marker | Company | Catalog number | Application |
|---|---|---|---|
| Actin ExM 546 (phalloidin derivate) | Chrometra, Kortenaken, Belgium | n.a. | ExM |
| Cholera toxin subunit B (recombinant), Alexa Fluor 488 Conjugate | Thermo Fisher Scientific, Waltham, ME, USA | C34775 | ICC |
| Monoclonal mouse anti- protein gene product 9.5 | AbD serotec, Puchheim, Germany | 7863–1004 | srAT; ExM |
| Monoclonal rabbit anti-S100β | Abcam, Cambridge, UK | ab52642 | srAT |
| Mouse anti-Neurofilament marker (pan-neuronal, cocktail) | BioLegend, San Diego, CA, USA | 837801 | ICC |
| Polyclonal guinea pig anti-Desmoplakin | Progen, Heidelberg, Germany | DP-1 | srAT |
| Polyclonal rabbit anti- protein gene product 9.5 | Zytomed, Berlin, Germany | 516-3344 | ExM |
| Polyclonal rabbit anti-Connexin 43 | Sigma Aldrich, St. Louise, MO, USA | C6219 | srAT, ExM, ICC |
| Polyclonal guinea pig anti-Connexin 43 | alomone labs, Jerusalem, Israel | ACC-201-GP | ExM |
| Monoclonal mouse anti-Cytokeratin 10 | Abcam, Cambridge, UK | ab1421 | ICC |
| Polyclonal guinea pig anti Cytokeratin 14 | Progen, Heidelberg, Germany | GP14 | ICC |
| Polyclonal rabbit anti-Synaptophysin | Merck, Darmstadt, Germany | AB9272 | ICC |
| Wheat germ agglutinin, Alexa Fluor 647 conjugate | Thermo Fisher Scientific, Waltham, ME, USA | W32466 | ICC |
| Wheat germ agglutinin, CF633 conjugate | Biotium, Fremont, CA, USA | 29024 | ExM |
| Secondary antibodies | | | |
| Alexa Fluor 488 donkey anti-mouse | Dianova, Hamburg, Germany | 715-545-150 | srAT, ExM |
| Alexa Fluor 488 donkey anti-rabbit | Dianova, Hamburg, Germany | 711-545-152 | ExM |
| CF568 donkey anti-rabbit | Biotium, Fremont, CA, USA | 20098–1 | ICC |
| CF568 donkey anti-guinea pig | Biotium, Fremont, CA, USA | 20098–1 | ExM |
| CF633 goat anti-rabbit | Biotium, Fremont, CA, USA | 20122–1 | ExM, ICC |
| Cy3 goat anti-guinea pig | Dianova, Hamburg, Germany | 106-165-003 | srAT |
| SeTau-647 anti-Rabbit | Conjugated antibody kindly provided by Prof. Markus Sauer, Department of Biotechnology and Biophysics, University of Würzburg, Germany. | | srAT |

Abbreviations: ExM, expansion microscopy; ICC, immunohistochemistry; n.a., not applicable; srAT, super-resolution array tomography.

added to the lower side of the LR-White block prior to cutting. Ribbons were collected on poly-L-lysine coated slides (Thermo Fisher Scientific, Waltham, MA, USA).

## srAT immunolabeling, fluorescence imaging, and image processing

Primary and secondary antibodies used for srAT experiments are listed in *Table 1*. Ultrathin serial tissue sections were encircled via a pap pen (Science Services, München, Germany). A blocking solution containing 0.05% (v/v) tween20 and 0.1% (w/v) bovine serum albumin in PBS was added for 5 min. Primary antibodies diluted 1:400 in blocking solution were then dropped onto the slides while the initial solution was removed by applying filter paper on the adjacent side of the encircled area. Primary antibodies were incubated for 1 hr in closed humid chambers at room temperature (RT). Samples were washed four times with PBS in 5 min intervals. Afterwards, secondary antibodies were applied for 30 min at RT at 1:400 dilution in blocking solution containing 1:10,000 4',6-diamidino-2-phenylindole (DAPI; Sigma Aldrich, St. Louise, MO, USA) in the closed humid chamber.

Samples were washed again four times with PBS and a last washing step with double distilled H$_2$O (ddH$_2$O) for 5 min was added. Slides were dried with filter paper, mounted in mowiol 4–88 (Roth, Karlsruhe, Germany), and covered with high precision cover glass No. 1.5 H (Roth, Karlsruhe, Germany).

Image acquisition was performed via the Zeiss ELYRA S.1 SR-SIM with 63 x oil-immersion objective plan-apochromat 63 x, NA 1.4 Oil Dic M27 and ZEN (black edition) software (all Zeiss, Oberkochen, Germany) with PCO Edge 5.5 sCMOS camera (PCO, Kelheim, Germany), using three rotations. 700 nm z-stacks in 100 nm increments around the observed focal point of DAPI staining per section were imaged. Fluorescence images were processed via Image J (version 1.51 n, National Institute of Health, Bethesda, MD, USA).

Channels were assigned to a defined color code and minimum and maximum of the image histogram adjusted for each channel separately.

For each channel, the image slice with the brightest signal and best focus within the z-stack was determined separately to adjust for different light emission wavelength of fluorophores. Each channel was exported as a portable network graphic (png) format file.

## Electron microscopy sample preparation and imaging

For contrasting and carbon coating of the samples, cover glasses were removed and mowiol was washed out with ddH$_2$O and blow dried. The object glass area containing the ribbon was cut out with a diamond pen (Roth, Karlsruhe, Germany). A 2.5% (w/v) uranyl acetate (Merck, Darmstadt, Germany) in ethanol solution was dropped onto the sections and incubated for 15 min at RT. Sections were briefly washed in 100% ethanol, 50% (v/v) ethanol in ddH$_2$O, and 100% ddH$_2$0, followed by 10 min incubation at RT with 50% (v/v) lead citrate solution in decocted H$_2$O containing 80 mM lead citrate (Merck, Darmstadt, Germany) and 0.12 M trisodium citrate (AppliChem, Darmstadt, Germany) (*Reynolds, 1963*). After washing in ddH$_2$O, sections were dried and attached to specimen pin mounts via carbon conductive tape (Plano, Wetzlar, Germany). Conductive silver (Plano) was applied, connecting the glass with the edges of the holder. A 5 nm carbon coat was applied, using a CCU-010 carbon coating machine (Safematic, Bad Ragaz, Switzerland). Serial sections were imaged in a JSM-7500F field emission scanning electron microscope (SEM; JEOL, Tokyo, Japan) with an acceleration voltage of 5 kV, a probe current of 0.3 nA, and a working distance of 6.0 mm. At each region of interest (ROI), several images with increasing magnification were acquired.

## srAT image processing, correlation, and modelling

Montage and alignment of scanning electron microscopy (SEM) images were achieved via the ImageJ plugin TrakEM2 (version 1.0 a, 04.07.2012)(*Cardona et al., 2012*; *Schindelin et al., 2012*). Images corresponding to the same section at different magnifications were merged within one layer with least squares montage in similarity mode and an alignment error of 10–20 pixel. After each z-layer was positioned, serial 100 nm sections were orientated via align layers, using similarity as transformation mode and 20–100 pixel alignment error. The ROI in each layer was exported in a tagged image file (tif) format. To correlate immunofluorescence (IF) and SEM information, associated IF channel images and montaged SEM images were loaded into the vector graphics editor program Inkscape (version 0.92.3, 11.03.2018) and processed according to a standardized protocol (*Markert et al., 2017*). IF channel images were overlaid and linked, leaving only the DAPI channel visible as first image layer. Opacity of IF images was reduced and DAPI-labeled heterochromatin was used as an independent and unbiased landmark for correlation. Linked IF images were linearly transformed (rotation and resizing, but no distortions) to fit the cell nuclei orientation of the EM image. When adequate overlay was achieved, a rectangular area containing the ROI was extracted and each layer exported as a png file. Corresponding IF and EM images were then imported into the image editor GIMP2 (Version 2.10.0, 02.05.2018) for appropriate overlay and exported as png files. For tracing IENF in 3D, the open source software package IMOD was used (*Kremer et al., 1996*). Alternating 100 nm sections were imaged via IF and correlated with their corresponding EM images. Within the 100 nm stepwise srAT stack, the trajectory of an IENF was volumetrically reconstructed as extrapolated tubular structure. Its position was determined based on PGP9.5 localization available for every second section.

Furthermore, distinguishable electron density compared to keratinocyte cytoplasm and absence of desmosomes between adjacent keratinocytes were considered to identify the IENF in the EM context.

## Expansion microscopy of thin skin sections

PFA fixed 10 µm skin cryosections were blocked in 10% BSA(w/v) in PBS for 30 min and incubated with primary antibodies against 1:100 PGP9.5 (AbD serotec, Puchheim, Germany) and Cx43 (Merck, Darmstadt, Germany) in 0.1% (w/v) saponin and 1% (w/v) BSA in PBS over night at 4 °C. Applied antibodies are listed in *Table 1*. After washing with PBS, secondary antibodies were applied for 2 hr at RT with 1% BSA (w/v) in PBS. After washing, sections were covered with droplets of PBS and stored at 4 °C until further processing of samples. Expansion microscopy (ExM) was adapted from former published protocols (*Tillberg et al., 2016*; *Zhao et al., 2017*). Skin sections were incubated with PBS containing 0.1 mg/ml Acryloyl-X (Thermo Fisher Scientific, Waltham, MA, USA) in dimethyl-sulfoxide (Sigma-Aldrich, St. Louis, MO, USA) over night at RT. Afterwards, 33 nM expandable phal-loidin derivate Actin ExM 546 (Chrometra, Kortenaken, Belgium), labeling actin cytoskeleton, was applied in PBS with 1% (w/v) BSA and 0.1 mg/ml Acryloyl-X for 1 hr at RT. Subsequently, a monomer solution containing 8.625% (w/w) sodium acrylate , 2.5% (w/w) acrylamide, 0.15% (w/w) N,N'-methy-lenbisacrylamide, and 11.7% (w/w) sodium chloride (all Sigma-Aldrich, St. Louis, MO, USA) in PBS was added at 4 °C for 30 min. Gelation was performed after replacement with fresh monomer solution, additionally containing 0.2% (w/v) ammonium persulfate , 0.2% (v/v) tetramethylethylenediamine , and 0.01% 4-Hydroxy-TEMPO (w/v) (all Sigma-Aldrich, St. Louis, MO, USA).

Sections were first incubated at 4 °C for 30 min followed by 2 h at 37 °C Gelated samples were digested in 4 U/ml proteinase K buffer (New England Biolabs, Ipswitch, MA, USA) with 50 mM Tris pH 8.0 (Serva, Heidelberg, Germany), 50 mM EDTA (Sigma-Aldrich, St. Louis, MO, USA), 0.5% (v/v) Triton X-100 (Thermo Fisher Scientific, MA, USA) and 0.8 M guanidine HCl (Sigma-Aldrich, St. Louis, MO, USA) for 2 hr at 60 °C. Subsequently, gels were washed 10 min at RT with PBS, then with 1:2500 DAPI (Sigma-Aldrich, St. Louis, MO, USA) in PBS for 20 min at RT and again 10 min in BPS at RT. Gels were transfer into a dark petri dish with 100 x times final gel volume of sterile ddH$_2$O with a razor blade. Gels were expanded for at least 1 hr at RT before direct post-expansion imaging or storage at 4 °C.

Labeled sections were imaged both in pre-expansion and post-expansion state with an DMi8 inverse microscope via 20 x dry objective HC PL FLUOTAR L 20 x/0.40, 11506243, LAS X software, and DFC3000 G monochrome camera (all Leica Microsystems, Wetzlar, Germany) to determine the expansion factor via manual alignment in inkscape. Further, imaging was performed using the ELYRA S.1 SR-SIM with 63 x water-immersion objective C-Apochromat, 63x1.2 NA, 441777–9970 (all Zeiss, Oberkochen, Germany), and a PCO Edge 5.5 sCMOS camera (PCO, Kelheim, Germany). Gels were imaged inside poly-D lysine (Sigma-Aldrich, St. Louise, MO, USA) coated imaging chambers (Thermo Fisher Scientific, Waltham, ME, USA) to prevent drifting. Non-computed (widefield) images were used. Min/Max values were processed via ImageJ (version 1.51 n, National Institute of Health, Bethesda, MD, USA) for visualization.

## Quantitative expansion microscopy of thick skin sections

As a practical application of ensheathment and Cx43 plaque quantification, we adjusted our ExM protocol to ensure improved fluorescence signal and expansion homogeneity across epidermal sections in 20 µm skin sections obtained from patients with SFN and healthy controls. Ten µg/ml wheat germ agglutinin (WGA) coupled with CF633 was applied with 10% BSA (w/v) in PBS for 1 hr at RT prior to antibody labeling of skin sections to visualize membranes and extracellular space. Further, PGP9.5 (Zytomed, Berlin, Germany) and Cx43 (alomone labs, Jerusalem, Israel) antibodies were applied with 0.1% (v/v) Triton X-100 (Thermo Fisher Scientific, MA, USA) instead of saponin. Anchoring was performed with 0.1 mg/ml acryloyl-X in carbonate buffer (*Truckenbrodt et al., 2019*). Gelation was carried out at 4 °C over night prior to 2 hr incubation at 37 °C. Digestion was extended to twice 24 hr at 60 °C to prevent partial expansion and tissue rupture. Non expanded nuclei were imaged via a 20 x objective either prior to expansion or from adjacent 20 µm sections of the same biopsy. For post expansion nuclei, 20 µm z-stacks from ROIs were used as maximum projection. ImageJ and CellPro-filer pipelines were applied for size extraction (see *Supplementary file 2*). Whole epidermal scans post-expansion were acquired with a ThunderImager equipped with a K5 monochrome camera using a ×63 magnification objective (HC PL APO 63 x/1.20 W CORR CS2) at 2x2 binning and 1 µm z-spacing and processed via Thunder Software for digital clearing and mosaic merge at 10% overlap (all Leica Microsystems, Wetzlar, Germany). Finally, ROIs containing IENF were cropped and single channels exported. WGA channel images were pre-processed in ImageJ via BaSiC plugin (*Peng et al., 2017a*)

to adjust for uneven signal intensity distribution across epidermal layers. ROIs were segmented with a machine learning model (neighbors) trained on manual segmentation of five consecutive z-planes for each channel with Dragonfly software (Object Research Systems, Montréal, Canada).

Binary images were smoothed via erosion and dilation functions (see *Supplementary file 2*) in ImageJ (version 1.51 n, National Institute of Health, Bethesda, MD, USA). Via Dragonfly ORS, a 3D box was drawn to extract only pixel information from the relevant epidermal skin area to exclude remaining dermal fiber information and the stratum corneum. Afterwards, ensheathment pixel (px) ratio (*Equation 1*) and Cx43 contact px ratio (*Equation 2*) were determined.

$$ensheathment\ ratio = \frac{\left[ px_{(totalPGP9.5)} - px_{(PGP9.5+WGAcoloc)} \right]}{px_{(totalPGP9.5)}} \tag{1}$$

$$Cx43\ contact\ ratio = \frac{px_{(PGP9.5+Cx43+WGAcoloc)}}{px_{(totalPGP9.5)}} \tag{2}$$

Imaging, ROI selection, and manual segmentation training were performed in blinded manner.

## Fully human co-culture system

All applied human cell lines were derived in the lab and routinely screened for mycoplasma contamination via Venor GeM Classic (Minerva Biolabs GmbH, Berlin, Germany) according to the manufacturer's protocol. Already established human-induced pluripotent stem cells (iPSC) derived from fibroblasts were differentiated into thoroughly characterized sensory neurons of a healthy control cell line as previously described (*Klein et al., 2023*). Co-culture chambers (ibidi, Gräfelfing, Germany) were attached to 12 mm BioCoat Poly-D-Lysine/Laminin coverslips (Corning, New York, NY, USA). Both inner chambers were additionally coated with 1:50 matrigel Growth Factor Reduced (Corning, New York, NY, USA) at 37 °C for 30 min. Four-week-old neurons were splitted via TrypLE (Thermo Fisher Scientific, Waltham, MA, USA).

Conditioned neuronal medium (*Klein et al., 2023*) was removed prior to TrypLE treatment, filtrated via 0.2 µm syringe filters (Sarstedt, Nümbrecht, Germany) and kept at 37 °C. Neurons were resuspended in 70 µl of filtered conditioned neuronal medium, seeded into one chamber compartment, and acclimated for 1 week.

Healthy control-derived primary keratinocytes were cultured (*Karl et al., 2019*) and seeded into the corresponding compartment. The chamber insert barrier separating the associated compartments was removed after 24 hr and medium exchanged either with fresh keratinocyte medium, comprising of EpiLife Medium supplemented with 1% EpiLife defined growth supplement, and 1% pen/strep (all Thermo Fisher Scientific, Waltham, MA, USA) or stored conditioned neuronal medium. After 6 days, co-cultures were fixed with 4% PFA (v/v) in PBS $^{Ca++ / Mg++}$ (PBS$^{++}$) at RT for 15 min and washed three times for 5 min in PBS. Briefly, coverslips were treated with 10 µg/ml wheat germ agglutinin, conjugated with Alexa Fluor 647 (WGA-647; Thermo Fisher Scientific) in PBS for 10 min at RT and washed two times with PBS. Subsequently cells were blocked 30 min with 10% FCS (v/v) and 0.1% (w/v) saponin in PBS$^{++}$, then labeled either with 1:100 anti-Cx43 (Sigma Aldrich, St. Louise, MO, USA) or 1:100 anti-synaptophysin antibodies for visualization of contact sites over night at 4 °C. Antibody solution contained 10% FCS (v/v) and 0.1% saponin (w/v) in PBS. After washing, secondary antibodies, 1:10,000 DAPI, and 10 µg/ml cholera toxin subunit B, conjugated with Alexa Fluor 488 (Thermo Fisher Scientific) were applied for 30 min at RT in antibody solution without saponin. Coverslips were transferred onto object holders, embedded with mowiol 4–88 (Roth, Karlsruhe, Germany) and -stored at 4 °C until further processing. For keratinocyte characterization the same labeling protocol was used with antibodies against cytokeratin 10 and cytokeratin 14 (both 1:200).

## 2D co-culture live imaging and fluorescence microscopy

Before barrier removal, co-cultures were transferred into a Lab-Tek chamber system (Thermo Fisher Scientific, Waltham, MA, USA) with two-well compartments. The remaining well was filled with 500 µl PBS$^{++}$.

After barrier removal and addition of conditioned neuronal medium, co-cultures were incubated for 2d. For live-imaging, the Lab-Tek slide was transferred to an inverse DMi8 microscope operated on LAS X software (Leica Microsystems, Wetzlar, Germany) and equipped with a live imaging

chamber (ibidi, Gräfelfing, Germany). Phase contrast images with 20 x objective HC PL FLUOTAR L dry 20 x/0.40, 11506243 were taken in 20 min intervals for 67 h with a DMC2900 color camera (both Leica Microsystems, Wetzlar, Germany), with cells kept at 37 °C, 5% $CO_2$, and 20% $O_2$ (both v/v). Single regions were stitched as overview via ImageJ plugin MosaicJ (*Thévenaz and Unser, 2007*). Min/Max values were processed via ImageJ (version 1.51 n, National Institute of Health, Bethesda, MD, USA).

Fluorescently labeled co-cultures were imaged with an Axio Imager.M2 (Zeiss, Oberkochen, Germany), equipped with spinning disc-confocal system (X-light V1, CrestOptics, Rome Italy) and Spot Xplorer CCD camera (SPOT Imaging, Sterling Heights, MI, USA) operated on VisiView software (Visitron Systems, Puchheim, Germany) for overview. For super-resolution details, a Lattice-SIM with 63 x water immersion C-Apochromat 63 x/1.2 W Korr UV-VIS-IR M27, 21787-9971-790 and ZEN (black edition) software (all Zeiss, Oberkochen, Germany), with two aligned PCO Edge 4.2 M sCMOS cameras (PCO, Kelheim, Germany) was used. Min/Max values were processed via ImageJ (version 1.51 n, National Institute of Health, Bethesda, MD, USA). For 3D visualization, Min/Max values of complete z-stacks were adjusted per channel in ZEN (blue edition) software (Zeiss, Oberkochen, Germany) and depicted in 3D mode.

### Calcium imaging

First, customized well-chambers fitting three co-cultures per slide were assembled. Autoclaved high precision cover glass No. 1.5 H (Roth, Karlsruhe, Germany) were attached to the removable silicone chamber with three wells (80381, ibidi, Gräfelfing, Germany).

Wells were coated with poly-D lysine (Sigma-Aldrich, St. Louise, MO, USA) for 1 hr at 37 °C, followed by a washing step with sterile $H_2O$ and were subsequently dried. Finally, two well inserts (80209, ibidi, Gräfelfing, Germany) were placed in these wells and inner chambers additionally coated with 1:50 growth factor reduced Matrigel (Corning, Corning, New York, NY, USA) at 4 °C over night. Co-cultures were seeded as described above, with 20 ng/ml BDNF +20 ng/ml GDNF +20 ng/ml NGFb (all Peprotech, Rocky Hill, NJ, USA)+200 ng/ml ascorbic acid (Sigma-Aldrich, St. Louis, MO, USA) as supplement to the keratinocyte medium. At day 6, i.e. day after insert removal, medium was replaced by fresh keratinocyte medium containing 2 µM Fluo-8 AM (Abcam, Cambridge, UK) for 1 hr at 37 °C. Medium was changed to a live imaging solution consisting of pre-warmed FluoroBrite DMEM with 25 mM HEPES (both Thermo Fisher Scientific, Waltham, MA, USA). Cells were mounted on a ThunderImager, and imaged with a K5 monochrome camera at 2x2 binning with a HC PL FLUOTAR 40 x/0.80 PH2 (all Leica Microsystems, Wetzlar, Germany) after 15 min of acclimatization. One ROI per well of neurites in contact with keratinocytes was imaged at 4 Hz for 4 min at naïve state and relative fluorescence values were extracted via ImageJ and calculated as $\Delta F=F/F0$, with F0 as intensity at the first time point. Keratinocytes or neurites with at least 10% increase of $\Delta F$ compared to F0 were referred to as spontaneously active.

### Statistical analysis

Ensheathment and Cx43 plaques in clinical tissue sections of SFN patients versus healthy controls were compared via Mann-Whitney-U test. SPSS version 29 was applied for statistical testing (IBM, Armonk, NY, USA) and GraphPad Prism 9.4.1 (GraphPad Software, Inc, La Jolla, CA, USA) was used for data visualization.

### Acknowledgements

We thank Dr. Franziska Karl-Schöller for technical help in co-culture handling, Viktoria Diesendorf, BSc for routine sensory neuron differentiation, and Alexandra Gentschev, BSc for srAT test runs (all Department of Neurology, University Hospital of Würzburg, Germany). We also thank Daniela Bunsen, Claudia Gehrig-Höhn (Biocenter, Imaging Core Facility, University of Würzburg, Germany), and Dr. Sebastian Markert (Department of Cell Biology, Johns Hopkins University, Baltimore, USA) for expert technical help in srAT. We further thank Dr. Jan Schlegel (Department of Biotechnology and Biophysics, University of Würzburg, Germany) for the preparation of SeTau647 conjugated secondary antibody and Dr. Ralph Götz (Department of Biotechnology and Biophysics, University of Würzburg, Germany) for the introduction into expansion microscopy.

# Additional information

## Funding

| Funder | Grant reference number | Author |
|---|---|---|
| Deutsche Forschungsgemeinschaft | DFG UE171/4-1 | Nurcan Üçeyler |
| Deutsche Forschungsgemeinschaft | UE171/15-1 | Nurcan Üçeyler |
| European Research Council | ULTRARESOLUTION | Markus Sauer |

The funders had no role in study design, data collection, and interpretation, or the decision to submit the work for publication.

## Author contributions

Christoph Erbacher, Conceptualization, Formal analysis, Investigation, Visualization, Methodology, Writing – original draft; Sebastian Britz, Investigation, Methodology, Writing – original draft; Philine Dinkel, Investigation, Methodology; Thomas Klein, Methodology; Markus Sauer, Resources, Methodology; Christian Stigloher, Conceptualization, Methodology, Writing – original draft; Nurcan Üçeyler, Conceptualization, Resources, Supervision, Funding acquisition, Writing – original draft

## Author ORCIDs

Christoph Erbacher  http://orcid.org/0000-0001-5931-6673
Thomas Klein  https://orcid.org/0000-0002-2719-9617
Markus Sauer  http://orcid.org/0000-0002-1692-3219
Christian Stigloher  http://orcid.org/0000-0001-6941-2669
Nurcan Üçeyler  https://orcid.org/0000-0001-6973-6428

## Ethics

Written informed consent and consent to publish was obtained. Our study was approved by the Würzburg Medical School Ethics committee (#135/15).

## Decision letter and Author response

Decision letter https://doi.org/10.7554/eLife.77761.sa1
Author response https://doi.org/10.7554/eLife.77761.sa2

---

# Additional files

## Supplementary files

• Supplementary file 1. Clinical parameters for all participants assigned to applied biomaterial.

• Supplementary file 2. Fiji macros and CellProfiler pipelines applied for segmentation.

• Transparent reporting form

## Data availability

All data generated or analysed during this study are included in the manuscript and supporting file. Imaging source data files have been provided for Figure 1 and are available at Zenodo. Numerical data for Figure 5 are provided in supporting files Figure 5—source datas 1 and 2.

The following dataset was generated:

| Author(s) | Year | Dataset title | Dataset URL | Database and Identifier |
|---|---|---|---|---|
| Erbacher C, Britz S, Dinkel P, Klein T, Sauer M, Stigloher C, Üçeyler N | 2022 | Interaction of human keratinocytes and nerve fiber terminals at the neuro-cutaneous unit | https://doi.org/10.5281/zenodo.6090262 | Zenodo, 10.5281/zenodo.6090262 |

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
