## [Editor Report]

In this important study, Erbacher et al. have used new techniques to explore the neuro-cutaneous structures of human epidermis. Human skin is less studied than rodent skin because it presents challenges in obtaining samples and finding excellent immunohistological labels. Here, the authors have employed expansion microscopy and super-resolution array tomography for histological studies and have developed a human keratinocyte and human iPSC-derived sensory neuron co-culture to provide in vitro data from live cells. Together, the data are compelling, and demonstrate that human axons tunnel through keratinocytes where they form gap junctions that allow for direct cellular communication.

---

## [Decision Letter]

**Decision letter after peer review:**

Thank you for submitting your article "Interaction of human keratinocytes and nerve fiber terminals at the neuro-cutaneous unit" for consideration by *eLife*. Your article has been reviewed by 3 peer reviewers, and the evaluation has been overseen by a Reviewing Editor and Kenton Swartz as the Senior Editor. The following individuals involved in the review of your submission have agreed to reveal their identity: Kathryn M Albers (Reviewer #1); Theodore J J Price (Reviewer #3).

Essential revisions:

Generally the reviewers we positive about the technical approach and potential impact. The major experimental points raised that should be addressed are (1) use specific antibodies to distinguish between sensory and skin cell types, (2) quantify the histological data that is currently just described and, (3) provide direct evidence for the gap junctions beyond Cx43 expression and plaques in keratinocytes (which was previously known). The last point was deemed critical since clarifying the presence of gap junctions would be a significant advance

I have attached the reviewer's specific comments. Please provide a point-by-point response to each when resubmitting the work.

*Reviewer #1 (Recommendations for the authors):*

Very nice study – the video files were a real treat!

The term nociceptor is used to describe nerve fibers. Without physiological or immunochemical confirmation, this descriptor should not be applied. Nociceptor should be replaced with a general descriptor, e.g., nerve fiber, projection, afferent.

Page 4, line 78 – insert CLEM as an abbreviation for correlative light and electron microscopy.

Figure 1c' – In the legend, 'co' is stated as the abbreviation for collagen but it is not shown in the panel. In addition, 'c' is used in panel 1c'.

Figure 1c' – I am confused by the mitochondria label; it appears to be within the collagenous (acellular) region of the dermis.

Figure 3 – has Cx43 defined in the legend but Cx43 labeling is not used in the figure.

Somatosensorics is not a familiar term used in the field. The use of monopole in the first sentence is also unusual.

A statement on sample size estimation (as in Author guidelines) should be included.

Please include the age of the skin donors.

A prediction from this study could be that Cx43 labeling is at a higher density at sites of neural ensheathment. As connexin puncta are widespread in the membrane, one could argue that the noted association to regions of nerve fiber interactions occur by chance. To address this, a relative measure of Cx43 labeling density in regions with and without fiber association could be included.

Similarly, in co-cultures, what percentage of visualized fibers that extended to keratinocytes exhibit close apposition to connexin immunolabeled structures?

It would be very interesting to know the type of nerve fiber being analyzed, e.g., peptide-rich (TrpV1+) or peptide-poor (TrpV1-), for both in vivo and culture analyses.

Is there an update on the Klein et al. reference (p 12)? The protocol for differentiation should be included if not yet available.

P 18 – please define CSC, line 347 in text.

*Reviewer #2 (Recommendations for the authors):*

Throughout the paper, the staining would benefit from quantification. It is unclear how frequently the IENF are embedded in keratinocytes, or whether there is any particular enrichment of Cx43 near IENF in the skin samples or co-culture. This could be quantified.

Using orange, red and pink all in one image makes it difficult to distinguish colors, and I imagine this is worse for color blind individuals. More contrasting colors should be chosen for direct comparisons.

The continuity of a given labeled IENF through multiple sections is not very obvious- in video 1, for example, the PGP9.5 staining does not look continuous at all between frames. It casts some doubt as to how reliable this staining is.

The authors claim to distinguish Langerhans cells and dermal schwann cells using the same marker. While LCs might not be as common in the dermis, additional markers would be required to confidently identify these two cell types. Moreover, the labeled schwann cell shows very low immunoreactivity to S100B, but human SC's are robustly labeled with S100B antibodies (Stratton et al., 2017, eNeuro).

The authors do make some claims that their co-culture techniques would allow them to answer, such as "Ensheathment of fibers by keratinocytes may orchestrate nerve fiber outgrowth and stabilization." (Figure 9 legend). This is mere speculation as it stands, but their imaging system would allow them to quantify this potential stabilization.

*Reviewer #3 (Recommendations for the authors):*

I think this is a strong paper. I do think there are some weaknesses that should be addressed, and some questions that are outstanding that the authors might want to discuss:

Weaknesses:

1) I recognize that the methods used here are challenging, and only sample small areas of the skin, but it would be useful to understand to what extent these arrangements are seen in the skin. Do the authors have the ability to quantify whether most sensory axons form these arrangements or if they are only seen with a particular subset of neuron (because they are relatively rare)?

2) Along the same lines, PGP9.5 is not specific for a particular type of sensory neuron and is also found in sympathetics and other neurons. The in vitro experiment supports the idea that these are sensory neurons, but have the authors considered looking at a more specific marker of nociceptors, for instance, like TRPV1? This could be quite informative.

Outstanding questions:

1) The impact of the paper is good, as it stands, but it could be improved by demonstrating that the gap junction connections are functional. I expected that the in vitro experiments would include such a finding, and was surprised that this was not tested. I think that evidence along these lines would really improve the paper. Having said that, the paper is already strong so this is a matter of preference I suppose. I do feel like readers will be expecting that data.

2) It would really be nice to better understand the phenotype of these neurons that are tunneling keratinocytes. We have good data on markers for neuronal subsets in humans from the 2 sequencing papers that have now come out. I realize these papers are new, so it is likely too much to ask to do this now, but there should be some discussion.

3) The burning question, in my mind, is what happens in small fiber neuropathy. Given the rich history of this lab's work in that area, I am sure that is coming in future papers, but I felt the discussion did not go into much detail on this.

4) I do not think that the authors really discussed their findings versus those of Talagas et al. 2020 sufficiently. My impression is that the Talagas paper is claiming synaptic contacts, but the expansion microscopy shows that these are more like en passant axons that form gap junctions. Are the ensheathments really devoid of gap junctions as shown in the summary figure?

---

## [Author Response]

Essential revisions:Reviewer #1 (Recommendations for the authors):Very nice study – the video files were a real treat!

Thank you!

The term nociceptor is used to describe nerve fibers. Without physiological or immunochemical confirmation, this descriptor should not be applied. Nociceptor should be replaced with a general descriptor, e.g., nerve fiber, projection, afferent.

We followed the Reviewer`s suggestion and have edited our wording accordingly.

Page 4, line 78 – insert CLEM as an abbreviation for correlative light and electron microscopy.

Thank you, we have added the abbreviation.

Figure 1c' – In the legend, 'co' is stated as the abbreviation for collagen but it is not shown in the panel. In addition, 'c' is used in panel 1c'.

We have changed the legend within 1C to “c” for collagen fibers.

Figure 1c' – I am confused by the mitochondria label; it appears to be within the collagenous (acellular) region of the dermis.

The respective mitochondria annotation in C```` refers to mitochondria (dark electron dense roundish structures) located within the nerve fiber, indicated by the white arrowhead. The area shown appears to be part of a dermal papilla. However, the nerve fiber is in close vicinity to a keratinocyte and continues to project further between keratinocytes. It is therefore termed “intraepidermal nerve fiber”.

Figure 3 – has Cx43 defined in the legend but Cx43 labeling is not used in the figure.

We apologize for this oversight and have edited the Figure respectively. Please be aware that this is now Figure 2.

Somatosensorics is not a familiar term used in the field. The use of monopole in the first sentence is also unusual.

Indeed, “Translational Somatosensorics” is the name of our section at our Department. We have, however, followed the Reviewer`s suggestion and have re-worded the respective passages.

A statement on sample size estimation (as in Author guidelines) should be included.

Since we are using non-high-throughput methods and are providing proof-of-principle data, our study is restricted to selected cases. Hence, sample size estimation is not applicable. We have added this statement in the ‘Materials and methods’ section of our revised manuscript (please see page 34):

“Since we used non-high-throughput methods and provide proof-of-principle data, our study was restricted to selected cases and sample size estimation was not applicable.”

Please include the age of the skin donors.

We have included a Supplementary file 1, characterizing all study participants (age, sex, intraepidermal nerve fiber density), and have added a respective statement in the Participants section (please see page 33-34):

“Epidemiological parameters of all study participants are summarized in Supplementary file 1.”

A prediction from this study could be that Cx43 labeling is at a higher density at sites of neural ensheathment. As connexin puncta are widespread in the membrane, one could argue that the noted association to regions of nerve fiber interactions occur by chance. To address this, a relative measure of Cx43 labeling density in regions with and without fiber association could be included.

We have followed the Reviewer`s suggestion and have added data for relative quantification of Cx43 contacts in skin samples of healthy controls (n = 5) and patients with small fiber neuropathy (n = 4). We have added respective passages in the ‘Materials and methods’ (see page 43-44), ‘Results’ (see page 15-17), and ‘Discussion’ (see page 25-26) sections of our revised manuscript. Please also see Figure 5.

Similarly, in co-cultures, what percentage of visualized fibers that extended to keratinocytes exhibit close apposition to connexin immunolabeled structures?

Several technical limitations impeded a quantitative analysis of connexin contact sites:

First, the commercial polyclonal rabbit anti-connexin 43 antibody used in the study (C6219; Σ Aldrich, St. Louise, MO, USA) did not result in specific labeling in two new consecutively ordered vials/charges, for both skin sections and cultured cells. After evaluation of several other commercially available antibodies, we found a suitable replacement (ACC-201-GP; alomone labs, Jerusalem, Israel) working well in skin sections, however, showing a higher degree of unspecific labeling in cultures (see Author response image 1).

**Author response image 1. sa2fig1:** 

Second, deriving co-cultures (2 weeks iPSC preparation and differentiation, 6 weeks maturation, and 1 week co-cultivation) currently represents a time consuming bottleneck.Third, related to the cell culture, many variables from differentiation efficiency (variation between iPSC passages, media and supplement lots, matrigel lots, etc.), success in neuronal split (variable cluster sizes, neuronal density, adhesion strength of neurons), keratinocyte growth (passage- and density-dependent), and other unknown factors influence the outcome of the later co-culture regarding number of outgrowing neurites and contact to keratinocytes. An infeasible amount of co-cultures would be needed to compensate this variability. Alternatively, a highly standardized co-culture format would be necessary as indicated in the ‘Discussion section’ of our revised manuscript (please see page 28):

“Specialized microfluidic chambers with directional neurite outgrowth, coupled with perfusion systems are needed to validate and quantify various modes of keratinocyte-neurite interactions at higher-throughput combined with keratinocyte-specific stimulation or antagonists like the gap junction blocker carbenoxolone.”

It would be very interesting to know the type of nerve fiber being analyzed, e.g., peptide-rich (TrpV1+) or peptide-poor (TrpV1-), for both in vivo and culture analyses.

As a pilot experiment, we have evaluated the use of TRPV1 labeling (see Figure Supplement 3), however, we refrain from integration into the main experiments (see page 9).

“To discriminate IENF subtypes, we applied immunoreaction against TRPV1, however, we could not extract IENF-specific co-localization, also after expansion (Figure 2—figure supplement 1).”

Despite being a commonly reported marker in immune-labeling experiments, the current opinion in the field is that specific antibodies for TRPV1 for human samples are not available. The use of any commercial antibody might result in uncertain or even erroneous data. With this caveat in mind, TRPV1 punctae appeared to be localized mainly within keratinocyte cytoplasm and close to keratinocyte membrane.

Co-localization with nerve fibers was observable in the dermis, however, due to keratinocyte signal, our achieved imaging resolution did not permit any conclusion on IENF. The observed labeling pattern may indicate specific labeling together with additional unspecific binding. For precise analysis a validated TRPV1 antibody, e.g. via previous evaluation of labeling of iPSC-derived human nociceptors with TRPV1 and TRPV1 knockout line, would be necessary. We are not aware of a sufficiently tested antibody and have no human TRPV1 knockout line available.

Is there an update on the Klein et al. reference (p 12)? The protocol for differentiation should be included if not yet available.

The manuscript is currently under review and meanwhile available as preprint. We have updated the information accordingly (see page 44):

“Already established human induced pluripotent stem cells (iPSC) derived from fibroblasts were differentiated into thoroughly characterized sensory neurons of a healthy control cell line as previously described (Klein et al., 2023).”

P 18 – please define CSC, line 347 in text.

We apologize for the confusion and have defined CSC, which is cutaneous Schwann cell.

Reviewer #2 (Recommendations for the authors):Throughout the paper, the staining would benefit from quantification. It is unclear how frequently the IENF are embedded in keratinocytes, or whether there is any particular enrichment of Cx43 near IENF in the skin samples or co-culture. This could be quantified.

Please see comment two of reviewer 2 regarding Cx43 quantification. We also added pilot data for relative quantification of IENF ensheathment in skin samples of healthy controls (n = 5) and patients with small fiber neuropathy (n = 4). We have added respective passages in the ‘Materials and methods’ (see page 43-44), ‘Results’ (see page 15-17), and ‘Discussion’ (see page 25-26) sections of our revised manuscript. Please also see Figure 5.

Using orange, red and pink all in one image makes it difficult to distinguish colors, and I imagine this is worse for color blind individuals. More contrasting colors should be chosen for direct comparisons.

Thank you for raising this point. Our applied palettes (based on Okabe_Ito) try to accommodate for color-impaired vision on multi-color images (https://thenode.biologists.com/data-visualization-with-flying-colors/research/). However, this was not implemented in Figure 4 and was now revised accordingly.

The continuity of a given labeled IENF through multiple sections is not very obvious- in video 1, for example, the PGP9.5 staining does not look continuous at all between frames. It casts some doubt as to how reliable this staining is.

PGP9.5 as a marker is a cytosolic deubiquitinase. It is not a continuous structure in contrast to e.g. cytoskeletal markers like actin or tubulin. In addition, only the surface of each 100-nm LR-white embedded slices can be labeled by antibodies (due to the resin). Together this leads to an intermitted signal, which nevertheless can be traced over consecutive slices.

The authors claim to distinguish Langerhans cells and dermal schwann cells using the same marker. While LCs might not be as common in the dermis, additional markers would be required to confidently identify these two cell types. Moreover, the labeled schwann cell shows very low immunoreactivity to S100B, but human SC's are robustly labeled with S100B antibodies (Stratton et al., 2017, eNeuro).

Determination of cell types is possible via integration of S100B labeling, layer localization (epidermis/dermis), and observed SEM cellular morphology (Kruger *et al.*, 1981; Tobin, 2006). Since this Figure served mainly as an outlook showcase for future applications, it was transferred to the supplement. Regarding the labeling density and signal intensity we also refer to the previous comment.

The authors do make some claims that their co-culture techniques would allow them to answer, such as "Ensheathment of fibers by keratinocytes may orchestrate nerve fiber outgrowth and stabilization." (Figure 9 legend). This is mere speculation as it stands, but their imaging system would allow them to quantify this potential stabilization.

The role of nerve fiber ensheathment for nerve fiber branching and stabilization was highlighted in *D. melanogaster* (Jiang *et al.*, 2019; Tenenbaum *et al.*, 2017). Hence, given our data recognizing the same phenomenon of ensheathment in human epidermis, we believe it is a valid and hypothesis to assume a similar function. The Reviewer is correct that the imaging systems necessary for further analysis are available at site, however, establishing and applying the respective protocols (e.g. in vivo labeling and long-term imaging via fluorescent reporter) would be a whole new study.

Reviewer #3 (Recommendations for the authors):I think this is a strong paper. I do think there are some weaknesses that should be addressed, and some questions that are outstanding that the authors might want to discuss:Weaknesses:1) I recognize that the methods used here are challenging, and only sample small areas of the skin, but it would be useful to understand to what extent these arrangements are seen in the skin. Do the authors have the ability to quantify whether most sensory axons form these arrangements or if they are only seen with a particular subset of neuron (because they are relatively rare)?

Please see above. We now added further data to investigate the frequency of ensheathment. Please see new Figure 5.

2) Along the same lines, PGP9.5 is not specific for a particular type of sensory neuron and is also found in sympathetics and other neurons. The in vitro experiment supports the idea that these are sensory neurons, but have the authors considered looking at a more specific marker of nociceptors, for instance, like TRPV1? This could be quite informative.

For detailed molecular and electrophysiological characterization of our iPSC derived sensory neurons, we refer to the available preprint (Klein *et al.*, 2023).

In case of tissue sections, please note that there is no sympathetic subset of IENF. For TRPV1 we refer to the comment and answer above from reviewer 1 and Figure 2—figure supplement 1.

Outstanding questions:1) The impact of the paper is good, as it stands, but it could be improved by demonstrating that the gap junction connections are functional. I expected that the in vitro experiments would include such a finding, and was surprised that this was not tested. I think that evidence along these lines would really improve the paper. Having said that, the paper is already strong so this is a matter of preference I suppose. I do feel like readers will be expecting that data.

In a pilot experiment, we established a calcium imaging protocol for our co-culture system. We recorded spontaneous activity, but could not include consistent data on stimulation or blockage of gap junctions, due to technical limitations. We currently do not have access to a perfusion system allowing gentle addition or washing out of reagents (e.g. carbenoxolone), therefore one co-culture well could only be used for one measurement. The number of available co-cultures was in general limited as explained above for reviewer 2. At baseline we detected spontaneous calcium transients in keratinocytes and neurites (please see Figure 9 and page 23):

“ca^2+^ imaging of human co-cultured sensory neurons and keratinocytes revealed spontaneous activity in both cell types (4/7 wells). Keratinocytes showed slower and longer ca^2+^ peaks, while neurites exhibited a sharp rise (Figure 9; Video 6). In three wells with neurites passing along keratinocytes, an increase of keratinocyte ca^2+^ preceded neurite ca^2+^ peaks. Together, this indicates functional activity in our co-culture model, possibly coupled between both cell types.”

2) It would really be nice to better understand the phenotype of these neurons that are tunneling keratinocytes. We have good data on markers for neuronal subsets in humans from the 2 sequencing papers that have now come out. I realize these papers are new, so it is likely too much to ask to do this now, but there should be some discussion.

We indeed tried to include TRPV1 (ACC-030, alomone labs) as a known and GPX2 (HPA003545, Σ-Aldrich) as a potential novel marker of sensory neuron subsets (assumed A-LTMR low-threshold mechanosensory neurons) extracted from literature (Tavares-Ferreira *et al.*, 2022). TRPV1 signal was challenging to obtain from fibers after penetrating towards the epidermis (see comments above) and we did not detect a specific labeling of any GPX2 fibers co-localizing with PGP9.5 signal. A thorough evaluation of suitable antibodies against described potential markers, e.g. including Oncostatin M receptor and Somatostatin (assumed silent nociceptors) proposed in the mentioned published sequencing data should be applied in human skin samples. However, this is out of the scope of this manuscript.

3) The burning question, in my mind, is what happens in small fiber neuropathy. Given the rich history of this lab's work in that area, I am sure that is coming in future papers, but I felt the discussion did not go into much detail on this.

We agree with the Reviewer that quantification of morphological features at the neuro-cutaneous unit will give invaluable new insights in diseases affecting small fibers.

We faced several challenges to advance our proof of principle data towards clinical application. First, inclusion of thicker skin sections used in routine (usually 20-40 µm) did not properly expand with the current ExM protocol, since tissue sections are more complex than cultured cells and skin is an especially challenging tissue. We overcame this in the end with an adapted ExM protocol (see page 43-44).

Second, the imaging of expanded tissue sections leads to an exponential increase in imaging data up to terabytes. In our lab, we currently do not have a SOP for automated big data integration and image segmentation.

Still, as a pilot experiment, we aimed to compare a small group of small fiber neuropathy patients (n = 4) versus healthy controls (n = 5). We incorporated these data in Figure 5 (see also page 15-17). In our small cohort we did not observe differences in patients, yet of note these ensheathment and contact ratios seem to be relatively homogenous for controls and may be more variable in small fiber neuropathy patients. With the rise of cloud data storage and AI tools we are optimistic that skin sections from larger cohorts can be analyzed and interpreted at super-resolution in future, however this is beyond the scope of this manuscript.

4) I do not think that the authors really discussed their findings versus those of Talagas et al. 2020 sufficiently. My impression is that the Talagas paper is claiming synaptic contacts, but the expansion microscopy shows that these are more like en passant axons that form gap junctions.

We revised and extended our discussion on discrepancies and similarities of our findings and that of the Talagas group (Talagas *et al.*, 2020) in more detail in our revised manuscript. Please see page 27-28:

“Additionally, SYP, synaptotagmin, and syntax in 1A were successfully labeled together with cytokeratin 6 as a keratinocyte marker and pan-neurofilament as a neurite marker, suggesting en passant synapse-like contacts (Talagas et al., 2020b). Further, this study described tunneling fibers and “gutters” in their heterologous co-culture system, being congruent to our observations in the human keratinocyte- sensory neuron system. However, neurofilaments represent intermediate filaments within the cytoplasm and may not encompass the whole neuronal outline compared to membrane labeling (see Figure supplement 5). In accordance with findings from native DRG neurons, our sensory iPSC neurons contained SYP accumulations within the cytoplasm (Chou et al., 2002; Chung et al., 2019), deeming a membranous labeling necessary to clearly localize SYP in keratinocytes and rule out a neuronal localization within a co-culture approach. In our study, we did not observe specific SYP-positive clusters or vesicles in the cytosol of primary human keratinocytes associated with passing neurites. Still, electrophysiological activity of neurons in contact with keratinocytes has shown to be attenuated after blockage vesicular secretion of keratinocytes via botulinum neurotoxin type C, hinting towards a physiological role of synaptic release in signal transduction at the NCU (Talagas et al., 2020b).”

We believe that an even more extensive discussion solely on Talagas et al., 2020b would overburden the ‘Discussion’ section. In general, these colleagues also hypothesize an en-passant synapses, however, via chemical/vesicular release of signaling molecules. The groups shows presence of the mentioned vesicular proteins via Western blot and immunolabeling, connected to electrophysiology with and without vesicular blockage via botulinum toxin, which showed an effect on neuronal currents and conductance. Still our colleagues did not show or compare action potentials in this setup. From available sequencing data, a low expression of the investigated synaptic genes in keratinocytes is known (https://www.proteinatlas.org/). Taken together, en-passant vesicle release may represent one of several signaling routes. We caution on imaging/antibody-based conclusions on localization/structure in general and specifically in this case. Conversely, we focused on multiple imaging modalities (srAT, ExM, skin, co-culture) and selected markers (e.g. for membrane) to validate morphological findings in the human system, yet addressing the physiological axis less detailed, also due to already established literature regarding connexins (e.g. communication between keratinocytes and trigeminal neurons in co-culture attenuated by gap junction blocker carbenoxolone (Sondersorg *et al.*, 2014).

Are the ensheathments really devoid of gap junctions as shown in the summary figure?

Both in srAT and ExM, we did not observe Cx43 plaques at ensheathment sites.

References

Jiang, N., Rasmussen, J.P., Clanton, J.A., Rosenberg, M.F., Luedke, K.P., Cronan, M.R., Parker, E.D., Kim, H.-J., Vaughan, J.C., Sagasti, A., 2019. A conserved morphogenetic mechanism for epidermal ensheathment of nociceptive sensory neurites. *eLife* 8**,** e42455.

Klein, T., Gruener, J., Breyer, M., Schlegel, J., Schottmann, N.M., Hofmann, L., Gauss, K., Mease, R., Erbacher, C., Finke, L., 2023. Small fibre neuropathy in Fabry disease: a human-derived neuronal in vitro disease model. *bioRxiv***,** 2023.2008. 2009.552621.

Klusch, A., Ponce, L., Gorzelanny, C., Schafer, I., Schneider, S.W., Ringkamp, M., Holloschi, A., Schmelz, M., Hafner, M., Petersen, M., 2013. Coculture model of sensory neurites and keratinocytes to investigate functional interaction: chemical stimulation and atomic force microscope-transmitted mechanical stimulation combined with live-cell imaging. *J. Invest. Dermatol.* 133**,** 1387-1390.

Kruger, L., Perl, E., Sedivec, M., 1981. Fine structure of myelinated mechanical nociceptor endings in cat hairy skin. *J. Comp. Neurol.* 198**,** 137-154.

Mandadi, S., Sokabe, T., Shibasaki, K., Katanosaka, K., Mizuno, A., Moqrich, A., Patapoutian, A., Fukumi-Tominaga, T., Mizumura, K., Tominaga, M., 2009. TRPV3 in keratinocytes transmits temperature information to sensory neurons via ATP. *Pflugers. Arch.* 458**,** 1093-1102.

Sondersorg, A.C., Busse, D., Kyereme, J., Rothermel, M., Neufang, G., Gisselmann, G., Hatt, H., Conrad, H., 2014. Chemosensory information processing between keratinocytes and trigeminal neurons. *J. Biol. Chem.* 289**,** 17529-17540.

Talagas, M., Lebonvallet, N., Leschiera, R., Sinquin, G., Elies, P., Haftek, M., Pennec, J.P., Ressnikoff, D., La Padula, V., Le Garrec, R., 2020. Keratinocytes Communicate with Sensory Neurons via Synaptic‐like Contacts. *Ann. Neurol.* 88**,** 1205-1219.

Tavares-Ferreira, D., Shiers, S., Ray, P.R., Wangzhou, A., Jeevakumar, V., Sankaranarayanan, I., Cervantes, A.M., Reese, J.C., Chamessian, A., Copits, B.A., Dougherty, P.M., Gereau, R.W.t., Burton, M.D., Dussor, G., Price, T.J., 2022. Spatial transcriptomics of dorsal root ganglia identifies molecular signatures of human nociceptors. *Sci. Transl. Med.* 14**,** eabj8186.

Tenenbaum, C.M., Misra, M., Alizzi, R.A., Gavis, E.R., 2017. Enclosure of Dendrites by Epidermal Cells Restricts Branching and Permits Coordinated Development of Spatially Overlapping Sensory Neurons. *Cell Rep.* 20**,** 3043-3056.

Tobin, D.J., 2006. Biochemistry of human skin--our brain on the outside. *Chem. Soc. Rev.* 35**,** 52-67.